# Non-replicating adenovirus based Mayaro virus vaccine elicits protective immune responses and cross protects against other alphaviruses

John M. Powers[1,2], Nicole N. Haese[1], Michael Denton[1], Takeshi Ando[1], Craig Kreklywich[1], Kiley Bonin[1], Cassilyn E. Streblow[1], Nicholas Kreklywich[1], Patricia Smith[1], Rebecca Broeckel[1¤], Victor DeFilippis[1], Thomas E. Morrison[3], Mark T. Heise[4], Daniel N. Streblow[1,5]*

1 Vaccine and Gene Therapy Institute, Oregon Health and Science University, Beaverton, Oregon, United States of America, 2 Department of Molecular and Medical Genetics, Oregon Health and Science University, Portland, Oregon, United States of America, 3 Department of Immunology and Microbiology, University of Colorado School of Medicine, Aurora, Colorado, United States of America, 4 Department of Genetics, Department of Microbiology and Immunology, The University of North Carolina at Chapel Hill, Chapel Hill, North Carolina, United States of America, 5 Division of Pathobiology and Immunology, Oregon National Primate Research Center, Beaverton, Oregon, United States of America

¤ Current address: Rocky Mountain Laboratories, NIH/NIAID, Hamilton, Montana, United States of America
* streblow@ohsu.edu

**Data Availability Statement:** All relevant data are within the manuscript and its Supporting Information files.

## Abstract

Mayaro virus (MAYV) is an alphavirus endemic to South and Central America associated with sporadic outbreaks in humans. MAYV infection causes severe joint and muscle pain that can persist for weeks to months. Currently, there are no approved vaccines or therapeutics to prevent MAYV infection or treat the debilitating musculoskeletal inflammatory disease. In the current study, a prophylactic MAYV vaccine expressing the complete viral structural polyprotein was developed based on a non-replicating human adenovirus V (AdV) platform. Vaccination with AdV-MAYV elicited potent neutralizing antibodies that protected WT mice against MAYV challenge by preventing viremia, reducing viral dissemination to tissues and mitigating viral disease. The vaccine also prevented viral-mediated demise in IFNαR1-/- mice. Passive transfer of immune serum from vaccinated animals similarly prevented infection and disease in WT mice as well as virus-induced demise of IFNαR1-/- mice, indicating that antiviral antibodies are protective. Immunization with AdV-MAYV also generated cross-neutralizing antibodies against two related arthritogenic alphaviruses–chikungunya and Una viruses. These cross-neutralizing antibodies were protective against lethal infection in IFNαR1-/- mice following challenge with these heterotypic alphaviruses. These results indicate AdV-MAYV elicits protective immune responses with substantial cross-reactivity and protective efficacy against other arthritogenic alphaviruses. Our findings also highlight the potential for development of a multi-virus targeting vaccine against alphaviruses with endemic and epidemic potential in the Americas.

**Funding:** The work presented in this manuscript work was supported by grants from the National Institutes of Health 1U19AI109680-01 (DNS) and R41AI138964-01 (DNS). The funders had no role in study design, data collection and analysis, decision to publish, or preparation of the manuscript.

**Competing interests:** The authors have declared that no competing interests exist.

## Author summary

Mayaro virus is an understudied alphavirus that is currently circulating in tropical environments in South and Central America without an approved vaccine. Recent outbreaks have suggested a broadening range and higher likelihood of urban outbreaks, increasing the public health risk. Mayaro virus is closely related to other arthritogenic alphaviruses with overlapping circulation such as chikungunya and Una viruses, both of which also lack clinically approved vaccines. Identification of a safe, easily manufactured, and effective strategy to vaccinate at risk populations is important to control outbreak potential. Here we report on a vaccination approach using a non-replicating adenovirus viral vector that encodes Mayaro virus structural proteins that assemble into non-infectious virus-like particles upon expression following vaccination. These particles stimulate strong immune responses against Mayaro virus. Upon testing against other alphaviruses, it was determined that the vaccine elicits cross-reactive neutralizing antibodies against chikungunya and Una viruses, significantly diminishes disease severity, and protects immunocompromised, highly susceptible mice from death following viral challenge. Our study provides new approaches to protect against these co-circulating viruses using a single vaccine. This approach is highly amenable to other virus targets for vaccine development and its ability to provide protection against chikungunya virus has global ramifications.

## Introduction

Mayaro virus (MAYV) is a mosquito-transmitted alphavirus that circulates in zoonotic cycles in non-human primates, birds, and rodents with occasional spillover into human populations that can lead to urban spread [1]. The ability of the virus to infect both *Aedes* and *Culex* mosquitos and a wide range of vertebrate hosts potentially permitting both enzootic and urban transmission cycles [2]. MAYV is endemic to Central and South America and was first discovered in 1954 in Trinidad and Tobago [3]. Forest workers or visitors to forested areas are at increased risk of becoming infected. Upon returning to urban areas, this can lead to human outbreaks [3]. Human infection with MAYV leads to fever, myalgia, arthralgia, and rash, which are common symptoms of infection with other arthritogenic alphaviruses. MAYV febrile symptoms typically last for 3–5 days, although joint and muscle pain can persist for up to one year [2,3]. Based on similarity to other more prevalent alphaviruses, reduced reporting of MAYV infections could be due to misdiagnosis, most commonly as dengue fever or chikungunya disease [4].

The alphavirus genome is a positive single-stranded RNA approximately 11.5 kb in length that encodes 4 non-structural proteins (nsP1, 2, 3, 4) and 6 structural proteins (C, E3, E2, 6K, TF, E1). The structural proteins are translated as a single polyprotein from the subgenomic viral mRNA. First, the capsid protein (C) undergoes autoproteolytic cleavage, and the resultant C oligomerizes around the viral genome forming nucleocapsid structures. The remaining portion of the structural polyprotein is processed in the ER and cleaved into pE2 (E3-E2), 6K, and E1. E1 and pE2 form non-covalent heterodimers, and during trafficking through the Golgi secretory pathway pE2 is processed into E2 and E3 [5,6]. Processed glycoproteins are transported to the plasma membrane and encapsulated viral genomes are recruited for budding of viral particles. There are 3 genotypic strains of MAYV that have a narrow range of amino acid variability in the structural proteins. Genotype D is the most prevalent and viruses within this group have structural protein amino acid divergence of less than 3%. Slightly higher variability exists between genotypes L and D, although divergence is still less than 10% [7]. Such high amino acid similarity greatly increases the likelihood of shared antigenic domains, enabling a

vaccine to cross-protect against most, if not all, MAYV strains [3,7]. However, to date there are no approved alphavirus vaccines except an inactivated-virus vaccine for horses that is directed against Getah virus. Previous MAYV vaccination attempts have included live-attenuated virus, inactivated virus, chimpanzee adenovirus vectors, and DNA based vaccines [8–12]. Therapeutic approaches to limit disease severity have been another area of research interest. For example, the use of adenovirus vectors expressing an IFN-α transgene have shown efficacy in reducing the inflammatory response in mice challenged with CHIKV, indicating a role for adenovirus vectors as permissive approaches for therapeutics [13].

To this end, the development of a vaccine that elicits protective immunity against MAYV is of interest as recent studies have suggested that a number of mosquito species are capable of transmitting MAYV and that they have broadening distributions, thus increasing the potential for global spread of the virus to more distant geographical regions [14]. There are numerous vaccine platforms from which to choose when designing a MAYV vaccine including: Live-attenuated viruses (LAV), recombinant proteins, self-assembled virus-like particles (VLP), and other viral vectors. We chose to combine VLP and Adenovirus vectors as our approach to develop a MAYV vaccine since LAV alphavirus vaccines can be contraindicated in immune compromised individuals, such as the elderly, and recombinant protein vaccines are plagued by rapidly waning immunity [15–18]. Previous studies have shown that expression of full-length alphavirus structural proteins, either through direct DNA transfection of cells or by viral vectors, results in self-assembly of VLPs. These VLPs are structurally similar to native virus particles but are devoid of infectious viral genomes [19,20]. Adenovirus based vectors have previously shown vaccination potential against alphaviruses. Earlier studies discovered the potential of an AdV vaccine expressing CHIKV structural proteins that protected mice from disease following virulent CHIKV challenge [21–23]. Therefore, a recombinant adenovirus vector vaccine platform was utilized in the current study because adenovirus vectors have the capacity to accept the entire structural protein from MAYV. They also have considerable efficacy as gene therapy vectors due to their ability to stimulate both innate and adaptive immune responses by high transgene expression [24,25]. As outbreaks of these related alphaviruses have occurred in the same regions due to their overlapping circulation, a multivalent vaccine would be of great benefit to the region [26–28]. Previous studies have indicated the existence of conserved antigenic epitopes between these viruses, with groups reporting on the presence of cross-neutralizing antibodies following infection with CHIKV or MAYV in rhesus macaques, and that humans infected with CHIKV possessed antibodies against CHIKV, MAYV, and Una virus (UNAV) [29–31]. Similar data on the ability of alphavirus vaccines to cross-protect against heterotypic viruses has also been noted in mouse models [32–36]. Therefore, in addition to evaluating the efficacy of AdV-MAYV against MAYV, we also cross-examined its vaccine potency against CHIKV and UNAV, two related members of the Semliki Forest complex [37].

## Materials and methods

### Ethics statement

Mouse experiments were performed in an ABSL3 laboratory, accredited by the Association for Accreditation and Assessment of Laboratory Animal Care (AALAC) International, in compliance with animal protocols approved by the OHSU IACUC Committee.

### Cells

Vero cells (ATCC) and 293-IQ cells (Microbix; HEK293 cells expressing the lac repressor [38]) were propagated at 37˚C with 5% $CO_2$ atmosphere in Dulbecco's Modified Eagle Medium

(DMEM) supplemented with 5 or 10% Fetal Bovine Serum (FBS) and Penicillin-Streptomy-cin-L-Glutamine (PSG). Telomerized human fibroblasts stably expressing the coxsackie and adenovirus receptor (THF-CAR) were propagated in DMEM containing 10% FBS plus PSG. *Aedes albopictus* cells (C6/36 cells; ATCC CRL-1660) were propagated at 28˚C with 5% $CO_2$ in DMEM supplemented with 10% FBS and PSG.

## Viruses

Mayaro virus CH was generated from an infectious clone received from Dr. Thomas Morrison (UC-Denver). The following reagents were obtained through BEI Resources, NIAID, NIH as part of the WRCEVA program: Mayaro virus, Guyane, NR-49911; Mayaro virus, TRVL 4675, NR-49913; Mayaro virus, BeAr505411, NR-49910; Mayaro virus, Uruma, NR-49914; Una virus, MAC 150, NR-49912. CHIKV SL15649 and vaccine strain CHIKV 181/25 were gener-ated from their respective infectious clones as previously described [39]. Alphaviruses were grown in C6/36 cells. Viral stocks were prepared from clarified supernatants at 72 h post infec-tion (hpi) by ultracentrifugation over 10% sucrose (SW32Ti, 70 min at 76,755 x g). The virus pellets were resuspended in PBS and stored at -80˚C. Viral limiting dilution plaque assays using Vero cells were performed on 10-fold serial dilutions of virus stocks or tissue homoge-nates. The infected cells were continuously rocked in an incubator at 37˚C for 2 h, and then DMEM containing 5% FBS, PSG, 0.3% high viscosity carboxymethyl cellulose (CMC) (Sigma) and 0.3% low viscosity CMC (Sigma) was added to the cells. At 2 dpi, cells were fixed with 3.7% formaldehyde (Fisher) and stained with 0.2% methylene blue (Fisher). Plaques were visu-alized under a light microscope and counted.

## Adenovirus vaccine vector

A replication-incompetent human Ad5 adenovirus vector (containing E1 and E3 deletions) expressing the MAYV structural polyprotein (Capsid, E3, E2, 6K/TF, E1) was generated using the AdMax HiIQ system (Microbix). Briefly, the structural gene from MAYV BeAr505411 was cloned into pDC316(io) by first amplifying the gene by PCR with forward (ATATGAATTCA TGGACTTCCTACCAACTCAAGTG) and reverse (ATATAAGCTTTTACCTTCTCAAAG TCACACAAG) primers containing EcoRI and HindIII restriction sites, respectively. Resulting clones were sequence verified. For adenovirus rescue, 293-IQ cells were co-transfected with pDC316(io)-MAYVsp and pBHGloxΔE1,3Cre plasmid using Lipofectamine 2000 [38–40]. A modified Cytomegalovirus immediate-early promoter containing a lac repressor binding site inserted between the promoter and the open reading frame was used to drive transgene expression in cells lacking the Lac repressor. Adenovirus containing supernatants were col-lected at maximum cytopathic effect (CPE) and the virus vectors were passaged a total of four times in 293-IQ cells with the final production from the clarified supernatants of eleven infected T175 flasks. Virus was pelleted by ultracentrifugation at 79,520 x g for 70 minutes. The pellets were resuspended in a total of 1.5 ml of phosphate buffered saline (PBS) and stored at -80˚C. Adenovirus stocks were titered by limiting dilution CPE assay on 293-IQ cells in 96 well plates.

## Mouse experiments

IFNαR1$^{-/-}$ and WT C57BL/6N mice were housed in ventilated racks with free access to food and water and maintained on a 12 h light/dark cycle. WT C57BL/6N mice were purchased from Jackson Laboratories and IFNαR1$^{-/-}$ were used from an established breeding colony at the VGTI/OHSU. Animals were vaccinated with 100 μl AdV-MAYV (1x10$^6$ to 1x10$^8$ plaque forming units (PFU)) diluted in PBS injected into the posterior thigh muscle with or without a

boost at 14 days post-prime using the same viral dose. At times listed in the figure legends, blood was collected from the facial vein and allowed to clot before centrifuging for 10 minutes at 9,391 x g in a microcentrifuge in order to collect serum. Vaccinated mice were challenged with virus at either 28- or 84-days post-vaccination (dpv). A few groups of animals received passive transfer of 200 μl of pooled serum from five AdV-MAYV or mock vaccinated mice by intraperitoneal injection at 24 h before infection. Mice were challenged with $10^4$ PFU of $MAYV_{BeAr}$, $UNV_{MAC\ 150}$, or $10^3$ PFU $CHIKV_{SL15649}$, via a 20 μl injection into the right posterior footpad. Footpad swelling measurements were performed with digital calipers and health and weight were monitored daily following challenge. Serum was collected at 2 days post-infection (dpi) to measure viremia by plaque assay. These challenge studies were terminated at either 4 or 7 dpi at which time ankle, calf muscle, quadricep muscle, spleen, and serum were collected for use in plaque titration assays and qRT-PCR as previously described [39]. Tissue samples were collected into tubes with 0.5 ml of PBS and 2 mm beads (Propper Manufacturing Co., Inc.) for homogenization by bead beating using a Precellys 24 homogenizer (3 times x 45 seconds; Bertin Technologies). Cellular debris were pelleted and clarified lysate was used for infectious virus titration and cytokine multiplex assays. An additional 100 μl aliquot of the tissue homogenate was added to Trizol for qRT-PCR analysis. Histological analysis was performed on groups of five-week-old female WT C57BL/6N mice that received passive transfer of naïve or immune serum from vaccinated animals. One day after transfer, animals were challenged with $10^4$ PFU $MAYV_{BeAr}$ in the right hind limb footpad. A second group of naïve five-week-old female WT C57BL/6N were mock inoculated with PBS.

## Quantification of virus tissue burden

Tissues were harvested from infected mice into 500 μl PBS with approximately 20 glass beads in 2 ml Starstedt screw cap tubes. Tissues were bead beat in three 45 second cycles. Blood was collected and serum was collected from the clotted blood sample. A 20 μl sample of tissue lysates or sera were serially diluted in DMEM containing 5% FBS and PSG and added to 48-well plates containing a confluent monolayer of Vero cells and rocked at 37°C for 2 h. CMC in DMEM containing 5% FBS and PSG (250 μl per well) was overlaid and plates were incubated for 2 days at 37°C and then the cells were fixed with 3.7% formalin diluted in PBS and stained with 0.2% methyl blue dye for 15 minutes.

## qRT-PCR

A 100 μl aliquot of homogenized tissue was added to 900 μl Trizol and used for RNA extraction. cDNA was synthesized with superscript IV (Taqman qRT-PCR was performed on a QuantStudio 7 flex Real-Time PCR system. MAYV probe (TGGACACCGTTCGATAC) was used with forward (CCATGCCGTAACGATTG) and reverse (TACCACGGCCCGTCGGA CCTTC) primers, CHIKV probe (ACATACCAAGAGGCTGC) with forward (CCGTCCC TTTCCTGCTTAGC) and reverse (AAAGGTTGCTGCTCGTTCCA) primers, and UNAV probe (ACGGTACGCTTAAAAT) with forward (CGCGTTGGAGACGATCAGA) and reverse (TCCGATTTGGGCAGAGAACT) primers.

## Western blot analysis

THF-CAR were transfected with varying MOI's from 0 to 1000. Cells were harvested 72 hpi after washing with PBS and were pelleted at 391 x g for 15 minutes at 4°C in a refrigerated microcentrifuge. Pelleted cells were lysed in 300 μl cell lysis buffer for 30 minutes on ice. Lysed cells and debris were pelleted at 16,363 x g for 15 minutes in a microcentrifuge and supernatant was transferred to a new tube. 20 μl of lysate was run on a 4–12% Bis/Tris polyacrylamide

gel alongside MAYV infected Vero cells and control Vero cell lysate at 200 volts for 37 minutes. Semi-dry transfer was used to transfer proteins onto an activated PVDF membrane at 25 volts for 35 minutes. Membranes were blocked with 3% BSA/TBST, and probed with a 1:250 dilution of serum from $10^8$ prime + boost AdV-MAYV vaccinated mice. Monoclonal antibodies directed against CHIKV E1 and E2 (87.H1 and 133.B4) were kindly provided by Dr. Michael Diamond, Washington University at St. Louis. Anti-alphavirus capsid mAb has been previously described [41]. Membranes were then washed, and a secondary HRP conjugated rabbit α-Mouse IgG was used. Membranes were washed and then revealed with ThermoFisher Pico chemiluminescent developer solution and exposed onto X-ray film. $1.7x10^7$ PFU of $MAYV_{TrVl}$ was also run on a 4–12% Bis/Tris polyacrylamide gel, proteins were transferred, and membranes probed with a 1:250 dilution of serum from naïve or $10^8$ prime + boost AdV-MAYV following the above protocol.

## Electron microscopy

THF-CAR cells were infected with AdV-MAYV with a MOI = 100 PFU/cell. Media was harvested 72 hpi and cell debris was pelleted at 2,514 x g in a tabletop centrifuge for 10 minutes. Clarified media was then 0.22 μM filtered, and 10% Sorbitol was underlaid. Tubes were spun at 110,527 x g for 2 h. Supernatant was poured off and pellets were resuspended in 250 μl of PBS and 0.22 μM filtered. Resuspended samples were brought to a final volume of 1 ml in 15% trehalose [42], and a 50 μl sample was fixed in 4% PFA for 30 minutes and frozen at -80˚C. Samples were stained with uranyl acetate and EM images were taken by the OHSU Multiscale Microscopy Core on a Krios G4 Cryo-TEM.

## Neutralization assays

Blood was collected from mice and allowed to clot at room temperature for 30 minutes before centrifuging for 5 minutes at 3,000 x g. Sera was transferred to a new tube and heat inactivated at 56˚C for 30 minutes. A portion of sera was used for serial dilutions in DMEM supplemented with 5% FBS and 1% PSG. Diluted serum was mixed with media containing $MAYV_{CH}$, $MAYV_{BeAr}$, CHIKV 181/25, or $Una_{Mac150}$. Media and virus were incubated for 2 h at 37˚C while rocking. Serum and virus containing media was transferred to confluent 12 well plates of Vero cells and rocked for 2 h in a 37˚C 5% $CO_2$ atmosphere incubator. One milliliter of CMC DMEM medium containing 5% FBS was added to each well and the plate was incubated for 48 h in a 37˚C 5% $CO_2$ incubator. Plaques were fixed by adding 1 ml of 3.7% formaldehyde to each well for 15 minutes, then the supernatant was removed and the wells were washed with cold water and stained with 0.2% methyl blue dye for 15 minutes. Plates were washed with cold water to remove excess dye and dried prior to counting plaques. $PRNT_{50}$ was calculated by non-linear regression after determining the percent of plaques at each dilution relative to the average plaques in virus only control wells.

## Pre- and post-attachment neutralization assay

Two-fold serial dilutions of PBS or heat inactivated serum from PBS or AdV-MAYV vaccinated, as well as AdV-MAYV MAYV challenged mice were prepared in DMEM (1:80 to 1:81920). For pre-attachment assays, diluted serum was mixed with 11 PFU of $MAYV_{BeAr}$ and rocked for 1 h at 4˚C. Serum-virus complexes were then added to confluent 12-well plates of Vero cells and rocked for 1 h at 4˚C. Non-adsorbed complexes were removed by 3 washes of DMEM and plates were moved to a 37˚C incubator for 15 minutes to allow for internalization of bound virus. Plates were then overlaid with 1 ml of CMC DMEM medium containing 5% FBS. Post-attachment assays were conducted in a similar manner, but initially 11 PFU of

$MAYV_{BeAr}$ was added to 12-well plates and plates were rocked at 4˚C for 1 h. After washing to remove unbound virus, serum dilutions were added to the wells. Plates were again rocked for 1 h at 4˚C, followed by washing, and plates were moved to a 37˚C incubator for 15 minutes and overlaid with 1 ml of CMC DMEM medium containing 5% FBS. At 48 hpi, cells were fixed with 3.7% formaldehyde, stained with 0.5% methylene blue, and virus plaques were counted.

## Enzyme-linked immunoassay (ELISA)

$MAYV_{BeAr}$ was heat inactivated at 56˚C for 30 minutes and diluted in PBS to $9.4x10^4$ PFU/100 µl was added to each well of high binding flat bottom 96 well plates (Corning). Plates were sealed and incubated at 4˚C for 4 days to allow for virus coating. Plates were blotted dry and blocked for 1 h with 100 µl of 5% milk in PBS with 0.05% Tween-20 (ELISA buffer). Prior to use the plates were washed three times with 100 µl of ELISA buffer. Heat-inactivated sera from mice was diluted 1:50 in ELISA buffer and serially diluted 1:3 for a total of 6 dilutions. 100 µl of each dilution was added to wells and incubated for 1.5 h at room temperature. Plates were washed three times with PBS with 0.05% Tween-20 (ELISA wash buffer) and blotted dry. Secondary antibodies (goat anti-mouse IgG1 HRP conjugated–Rockland 610-103-040; goat anti-mouse IgG2b HRP conjugated–Rockland 610-103-042; goat anti-mouse IgG3 HRP conjugated–Rockland 610-103-043; goat anti-mouse IgG + IgM HRP conjugated–Rockland 610-103-115) were diluted 1:10,000 in ELISA buffer and 100 µl was added to wells and incubated for 1 h at RT. Plates were then washed three times with ELISA buffer, blotted dry and developed with OPD substrate. The reaction was stopped with 1 M HCl 10 minutes after exposure. Plates were read at 490 nm on a BioTek plate reader. Biological replicates were used to evaluate treatment groups.

## Enzyme-linked immunospot assay (ELISpot)

ELISpot assays were performed as previously described [39]. A single-cell suspension was created by grinding a whole spleen through a 70 µm cell strainer with the rubber end of a 3 mL syringe plunger and rinsing with 15 ml of RPMI with 10% FBS and 1% PSG (RPMI complete). Cells were pelleted at 650 x g for 10 minutes and red blood cells were lysed with 1x Red Blood Cell Lysis Buffer (Biolegend) for 3 minutes, after which 10 ml of complete RPMI was added and cells were pelleted. Cells were resuspended in 3 ml RPMI complete medium and counted. Splenocytes, $2.5x10^5$ cells per well, were added to Mouse IFN-γ ELISpot plates (MabTech) in addition to 20 µl of peptide (2 µg/well), 2 µl of DMSO, or 2 µl of a phorbol 12-myristate 13-acetate/ionomycin stock at a 1:300 dilution as a positive control. 18mer peptides corresponding to predicted H2b epitopes present in the MAYV structural polypeptide were ordered from Thermo Scientific. Plates were incubated for 48 h, washed and incubated with anti-mouse IFN-γ biotin antibody for 2 h. Plates were again washed and incubated with streptavidin-ALP antibody for 1 h following the manufacturers protocol. Spots were visualized using BCIP/NPT-plus substrate, after which the plates were washed and dried prior to counting with the aid of the AID EliSpot Reader Classic.

## Cytokine and chemokine analysis

A Milliplex MAP Mouse Cytokine Magnetic Bead Panel multiplex assay (Millipore Sigma) was used to detect 26 cytokines, chemokines, and growth factors in mouse tissue homogenates from vaccinated and control mice at 4 days post footpad $MAYV_{BeAr}$ challenge. Cytokines from tissues of AdV-MAYV and AdV-GFP vaccinated mice were analyzed using a R&D Systems mouse magnetic Luminex LXSANSN-26 assay. Briefly, 25 µl of clarified tissue homogenate from ipsilateral and contralateral ankles and quadriceps (N = 10 per group), calf (N = 8 per

group), and naïve mouse calf and ankles (N = 4 per tissue). The manufacturer's protocol was followed with minor alterations as previously described [39]. The plate was read on a Luminex 200 Detection system (Luminex).

## Histopathology

At 7 dpi, MAYV-infected mice were sacrificed and perfused with 4% paraformaldehyde in PBS. The lower hind legs were collected, fixed in 4% paraformaldehyde, decalcified, and embedded in paraffin, and then 5-μm sections were prepared. Mounted sections of ipsilateral and contralateral legs were stained with H&E and evaluated for inflammation and tissue disease by light microscopy (Olympus VS120 Virtual Slide Microscope). Anatomic pathology specialists blindly scored the presence, distribution and severity of histological lesions including necrosis, inflammation, fibrosis, edema, and vasculitis using a scoring system of 0–5 using the no infection control tissues as a baseline score: 0 absent (no lesions), 1 minimal (1~10% of tissues affected), 2 mild (11~25% affected), 3 moderate (26~50% affected), 4 marked (51~75% affected), 5 severe (>75% affected). All data were analyzed using GraphPad Prism 8 software.

## Statistical analysis

Statistics and graphs were created with GraphPad Prism 8. Normalized variable slope non-linear regression using upper and lower limits of 100 and 0, respectively, was used to calculate neutralizing antibody titers. T tests were used to compare normally distributed pairwise data sets and Mann-Whitney was used for pairwise comparison on non-normally distributed and/or when data points were below the assays limit of detection. ANOVA was used for data sets with three or more groups with normally distributed data and Kruskal-Wallis ANOVA was used for data sets of three or more groups with non-normally distributed and/or data points that were below the assays limit of detection.

# Results

## Adenovirus mediated expression of Mayaro structural proteins

The Mayaro virus (MAYV) structural protein ORF from a Buenos Aires isolate (BeAr505411) was cloned into a replication-defective adenovirus expression vector (AdV-MAYV) as previously described [39]. The CMV-IE promoter drives expression of the structural ORF (**Fig 1A**). Western blotting was used to confirm expression of the MAYV ORF in lysates from THF-CAR cells infected with a range of MOI's ranging from 0.1 to 1,000 PFU/cell. Lysates from uninfected and MAYV-infected Vero cells were included as negative and positive controls, respectively. Robust protein expression was observed for the vector in a dose-dependent manner. Blotting using an anti-alphavirus capsid antibody detected a band at ~30 kDa, which is the expected size of processed capsid protein [43] but we also expect that (**Fig 1B**). To determine whether transduction of cells with AdV-MAYV resulted in the production of intact virus-like particles (VLPs), THF-CAR cells were infected with a multiplicity of infection (MOI) equal to 100 PFU/cell and VLPs were purified from clarified supernatants by ultracentrifugation [19]. Resuspended VLPs were fixed with 4% PFA, counterstained with uranyl acetate and EM analysis identified particles of ~60–70 nm in size with the correct alphavirus morphology indicating that transduction of cells with AdV-MAYV generated virus-like particles [44](**Fig 1C**).

## AdV-MAYV vaccination elicits neutralizing antibodies and T cell responses

To determine optimal dose and dosing regimen, WT C57BL/6N mice were vaccinated intramuscularly (n = 5 per group) with single doses of $10^6$, $10^7$, or $10^8$ PFU AdV-MAYV. An

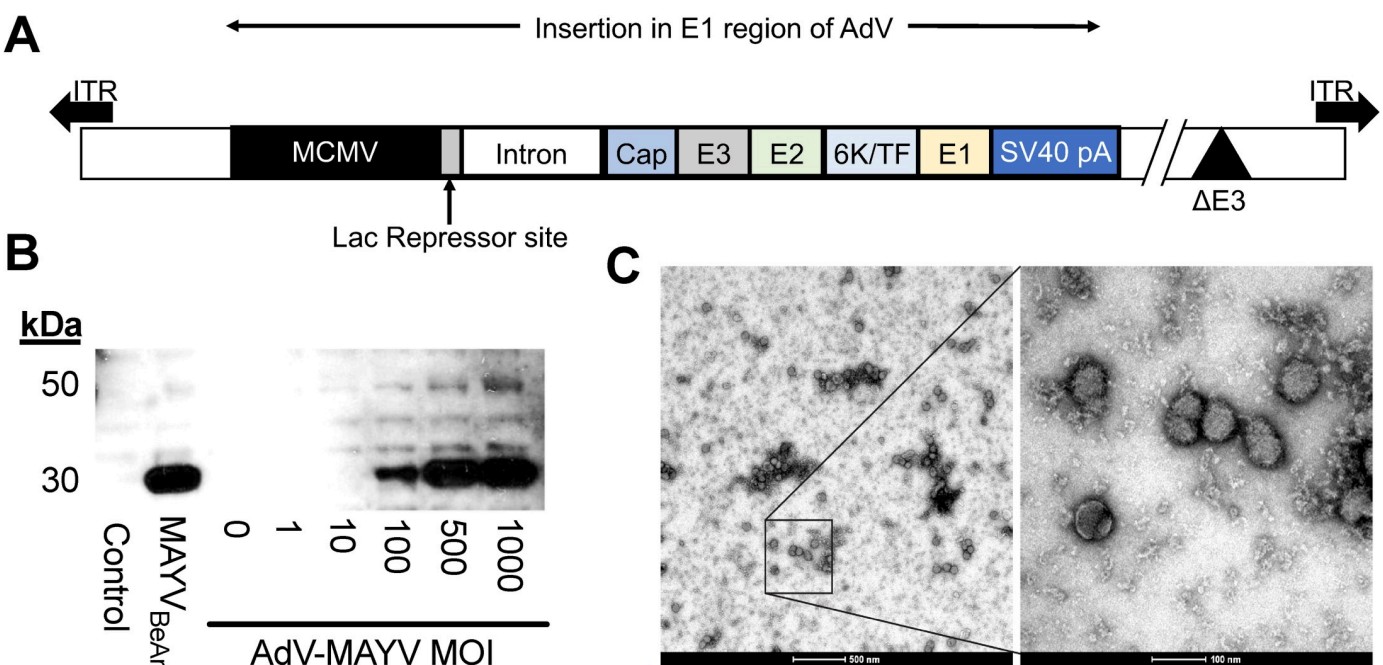

**Fig 1. AdV-MAYV Vaccine Vector Expression of MAYV Structural Proteins.** (A) The full length MAYV structural protein ORF was inserted into the replication defective huAdV expression vector under the control of the MCMV-IE2 promoter containing a Lac repressor to specifically modulate expression during reconstitution and stock production of the virus. (B) AdV-MAYV expression of the structural protein was evaluated in THF-CAR cells infected with increasing MOIs. Cell lysates were analyzed at 72 hpi for capsid expression by western blotting for the viral capsid protein. Shown is a representative image of two independent experiments. (C) Electron microscopy image of 4% PFA fixed AdV-MAYV VLPs from clarified supernatants of AdV-MAYV infected THF-CAR cells (MOI = 100 PFU/cell). Representative images are shown of 500 nm (left) and 100 nm (right) magnifications. Shown is a representative image of three biological replicates.

additional cohort of animals for each group received a booster vaccination at 14 days post vaccination (dpv) with the same vaccine and at the same dosage. Negative control animals were vaccinated with an AdV-GFP vector, and positive control animals were infected in the footpad with $10^4$ PFU MAYV$_{BeAr}$. We did not observe any changes in mouse behavior or weight suggesting that the vaccine dosage and route do not have any major safety concerns. Serum was harvested and heat inactivated for use in plaque reduction neutralization assays (PRNT) with MAYV$_{CH}$ on Vero cells as indicated in the experimental timeline (**Fig 2A**). All animals that received the AdV-MAYV vaccination generated neutralizing antibodies to a higher degree than the AdV-GFP control group (P < 0.0001) (**Fig 2B**). Increasing the AdV-MAYV dose, as well as providing a boost at 14 dpv increased levels of neutralizing antibodies substantially. Only the prime + boost animals receiving the $10^8$ PFU dose of AdV-MAYV developed neutralizing antibody levels higher than those observed in the serum from MAYV-infected mice (PRNT$_{50}$ of 4,123 vs 853). A dose of $10^7$ PFU AdV-MAYV prime + boost provided near equivalent neutralizing antibody production relative to MAYV-infected mice (PRNT$_{50}$ of 795 vs 853) (**Fig 2B**).

Serum was tested to assess whether neutralization of infection occurred pre- or post-viral binding to the cell using modified neutralization assays. For this assay, serum neutralizing antibody levels from mice vaccinated with AdV-MAYV ($10^8$ prime + boost) was compared to serum from naïve, PBS control, or AdV-MAYV $10^8$ prime + boost + challenged mice. In pre-attachment assays, a series of diluted serum was mixed with MAYV and pre-incubated for 1 h and then the mixture was added to a confluent monolayer of Vero cells to allow for plaque formation. For post-attachment assays, Vero cells were incubated with MAYV at 4°C to allow

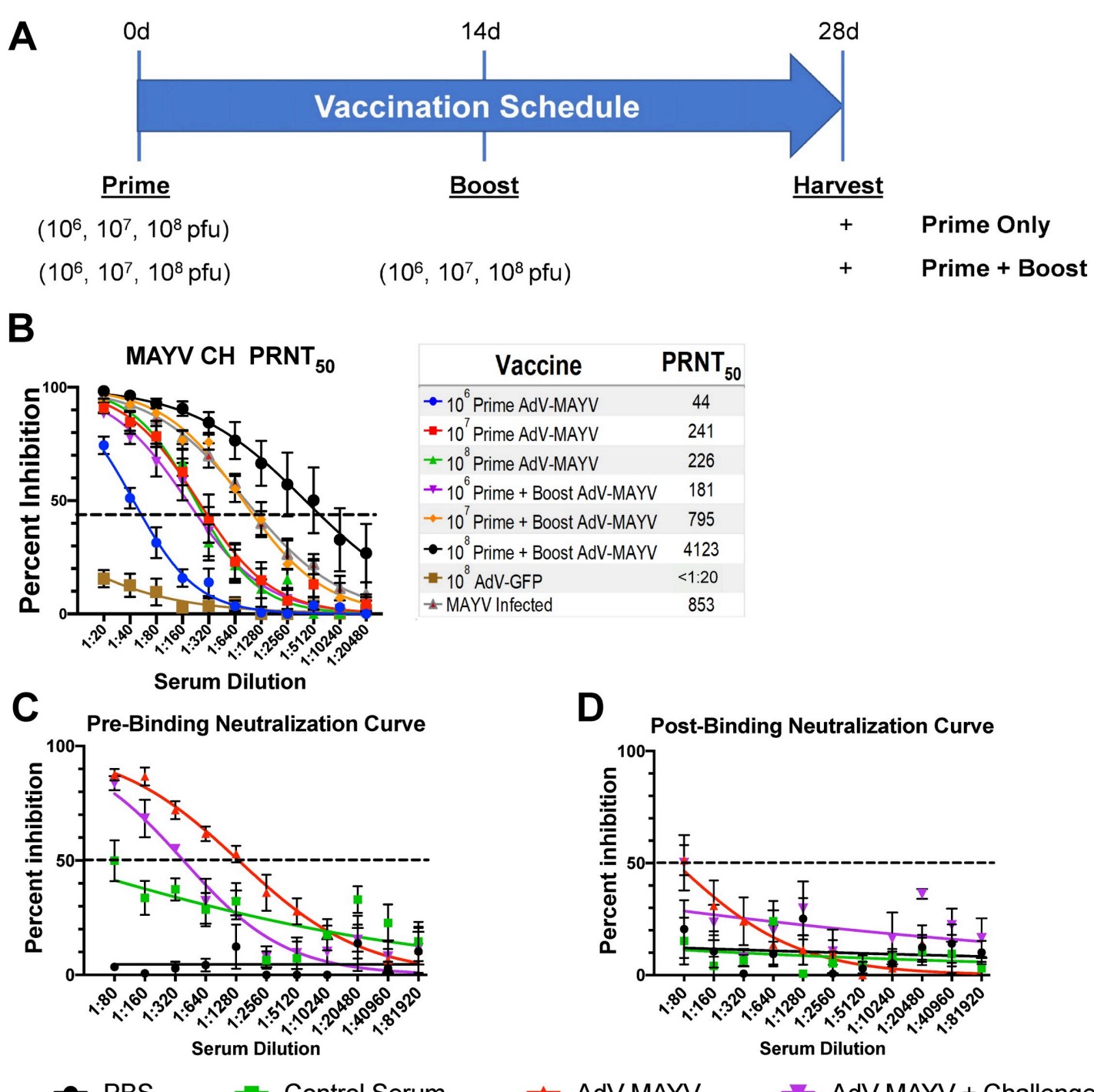

**Fig 2. AdV-MAYV-Induced Antibody Response Neutralize Virus in A Pre-attachment Mechanism.** (A) WT C57BL/6N mice were vaccinated with $10^6$, $10^7$, or $10^8$ PFU AdV-MAYV by intramuscular injection (n = 5 mice per group). A separate group of animals were infected with $10^4$ PFU MAYV$_{BeAr}$. At 14 days post vaccination, a subgroup of mice received a vaccine boost with the same vector and dosage as was used during the primary vaccination. Blood was collected from the vaccinated mice at day 28. (B) Sera from each mouse was tested for MAYV$_{CH}$ neutralization potential using a PRNT$_{50}$ assay. Shown are the average PRNT$_{50}$ values calculated for each group by non-linear regression at a 95% confidence interval (n = 5 mice per treatment group). PRNT50 values were calculated by variable slope non-linear regression. (C and D) Pre- and post-attachment neutralization assays were performed to explore the mechanism of inhibition. For pre-attachment neutralization assays, aliquots of a known concentration of virus were mixed with serial dilutions of serum for one hour prior to application to a confluent monolayer of Vero cells. For post-attachment treatments, virus was incubated with Vero cells at 4°C for one hour to allow binding and then serial dilutions of antibody were added for one additional hour at 4°C. Triplicate biological replicates and representative curves determined by variable slope non-linear regression are shown. Error bars represent SEM representative of 4 biological replicates.

attachment prior to incubation with a similar dilution series of serum. The greatest amount of neutralization occurred for serum in the pre-binding assay, indicating that the neutralizing antibodies stimulated from vaccination functioned to prevent virus binding to the cell. This was observed in serum from both the AdV-MAYV $10^8$ prime + boost mice and AdV-MAYV $10^8$ prime + boost challenged mice with greater than an 80% reduction in plaques at a serum dilution of 1:80 and $PRNT_{50}$'s of 1358 & 385, respectively (**Fig 2C**). Comparatively, post-binding analysis identified minimal neutralization and plaque reduction, with serum from vaccinated animals presenting a $PRNT_{50}$ dilution of only 66, while no other samples had plaque reduction above 50% (**Fig 2D**).

Based on the presence of strongly neutralizing antibodies, we next used ELISA to evaluate serum from naïve, $MAYV_{BeAr}$ infected, AdV-MAYV prime or prime + boost vaccinated mice for the presence of binding antibodies. Inactivated $MAYV_{BeAr}$ was bound to 96-well high-binding ELISA plates. Serial dilutions of each serum were added to the plates and then probed with secondary antibodies directed against mouse IgG1, IgG2b, IgG3, or total IgG/M to determine the isotype of the MAYV specific antibody responses. Vaccinated mice and MAYV-infected mice showed elevated levels of antiviral antibodies for all isotypes compared to negative controls (**Fig 3A**). AdV-MAYV prime + boost resulted in the production of higher levels of total antiviral antibodies (IgG/M) relative to AdV-MAYV prime only, which is consistent with the increase levels of neutralizing antibodies following vaccination booster (**Fig 2B**). While AdV-MAYV prime + boost vaccinated animals had similar levels of antibody subclass responses compared to the MAYV-infected group, the total IgG/M response was significantly higher for the MAYV infected group (**Fig 3A**). Western blotting was used to confirm MAYV antigen specificity and binding of the vaccine-elicited antibodies (**Fig 3B, Panels 1–4**). Serum from non-vaccinated controls did not detect MAYV proteins (**Fig 3B, Panel 3**). In contrast, serum from vaccinated mice detected MAYV E1/E2 glycoproteins (**Fig 3B, Panel 4**) comparable to those detected by anti-CHIKV monoclonal antibodies 87.H1 and 133.B4 (**Fig 3B, Panels 1 and 2**). In addition, serum from vaccinated animals also bound to the MAYV capsid protein (**Fig 3B, Panel 4**).

We previously demonstrated the positive effect that pre-formed CD8+ T cell responses have on alphavirus disease, and therefore, we next determined whether AdV-MAYV also induced virus-specific T cell responses using methods previously described [39]. T cell IFNγ-ELISPOT assays were performed using peptides predicted to be MAYV T cell receptor epitopes (IEDB analysis resource) for C57BL/6 mice [45]. Lymphocytes were prepared from spleens harvested from WT and interferon alpha receptor knockout ($IFN\alpha R1^{-/-}$) mice vaccinated with AdV-MAYV. The cells were plated onto IFN-γ ELISpot plates in the presence of 18mer peptides derived from the MAYV structural proteins, DMSO, or the positive control PMA/Ionomycin. After 48 h, plates were stained for the presence of IFN-γ and spots counted using an automated ELISpot reader. A peptide derived from sequences present in the N-terminal domain of MAYV E2 (LAKCPPGEVISVSFV) stimulated the strongest T cell production of IFN-γ in mice vaccinated with AdV-MAYV (**S1 Fig**). This response was significantly increased in response to AdV-MAYV vaccination relative to DMSO control (**S1 Fig**). There was a slight increase in the T cell response present in vaccinated $IFN\alpha R1^{-/-}$ mice versus WT mice.

## AdV-MAYV vaccine elicits protective efficacy and reduces inflammatory chemokine production

To test the efficacy of AdV-MAYV against MAYV infection, vaccinated WT mice were challenged by footpad injection with $10^4$ PFU $MAYV_{BeAr}$ at 28 dpv (**Fig 4A**). A second group of vaccinated mice was assessed for vaccine durability by challenge at 84 dpv. Blood collected at

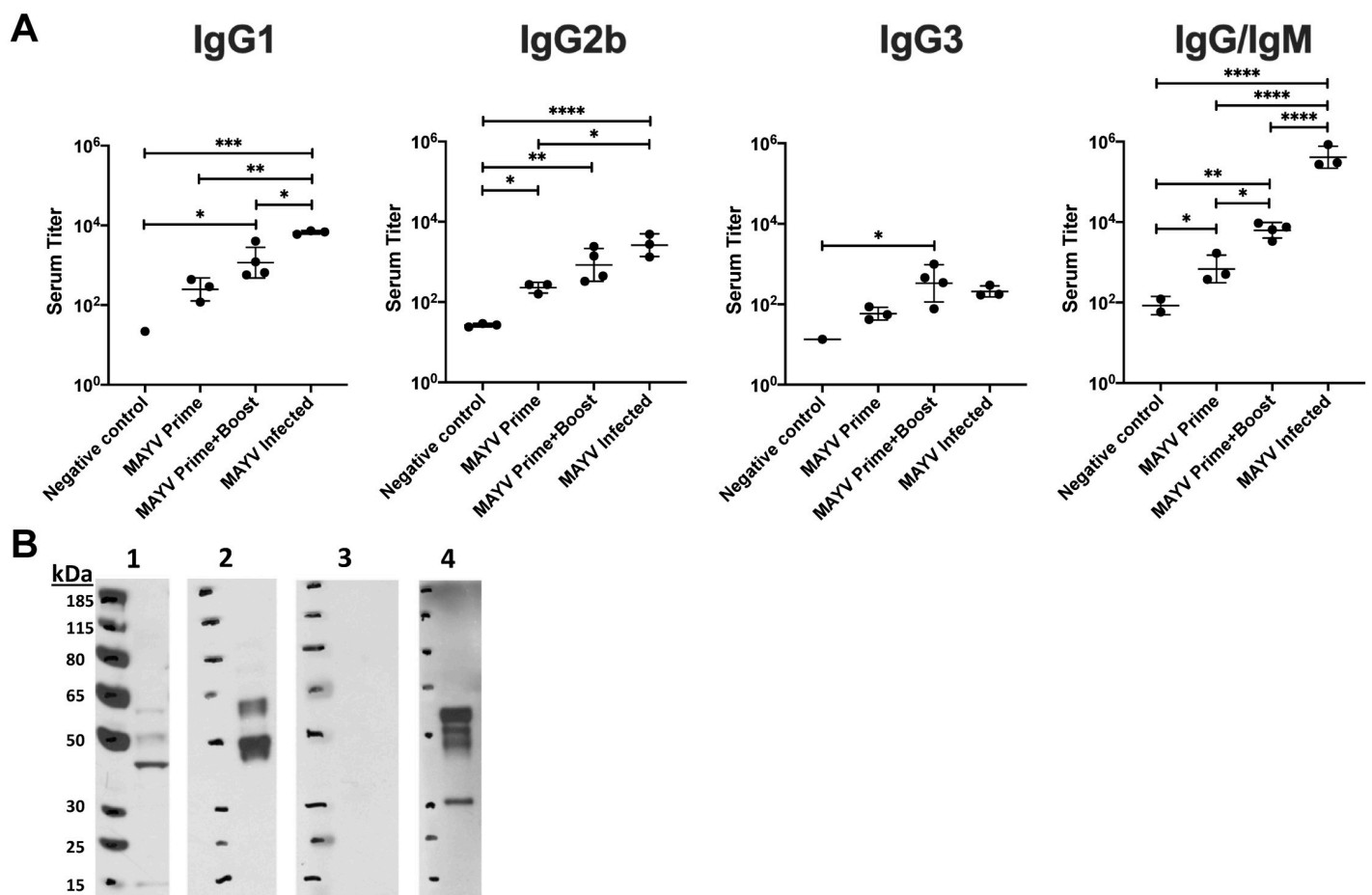

**Fig 3. Characterization of Anti-MAYV Antibody Responses.** (A) Isotype specific ELISAs were performed to characterize and measure the $MAYV_{BeAr}$ binding antibodies in sera from naïve, AdV-MAYV prime vaccinated, AdV-MAYV prime + boost vaccinated, and MAYV infected mice. Preparations of heat inactivated whole MAYV stocks were bound to high affinity 96-well plates. Serial dilutions of mouse sera were plated in order to calculate binding dilution titer. Binding antibodies were detected by secondary antibodies specific for mouse IgG1, IgG2a, and IgG3 as well as a pan IgG/IgM. Error bars represent SD representative of quadruplicate biological replicates. Statistical analysis was performed on log transformed data by a one-way ANOVA ($^*P < 0.05$, $^{**}P < 0.005$, $^{***}P = 0.0001$, $P < 0.0001$). (B) Western blot analysis was used to determine antigen specificity of antibodies in serum from vaccinated mice. Protein lysates containing purified $MAYV_{TrVl}$ were separated by SDS-PAGE and transferred to immunoassay membranes for western blotting. Cross-reactive anti-CHIKV E1 (panel 1) and E2 (panel 2) monoclonal antibodies were used to identify MAYV envelope proteins. Serum derived from naïve mice (panel 3) and AdV-MAYV prime + boost vaccinated mice (panel 4) demonstrate the presence of envelope and capsid specific antibodies to $MAYV_{TrVl}$ following vaccination. The blots shown are representative images of 3 independent experiments.

two days prior to challenge (26 and 82 dpv) displayed robust neutralizing antibody titers ($PRNT_{50}$ equal to 1,791 and 4,826; respectively) for AdV-MAYV vaccinated mice while control animals had no neutralization activity ($P < 0.0001$) (**Fig 4B**). After challenge, mice were monitored daily for footpad swelling and other signs of disease. At 2 dpi, blood was collected from the animals and sera was processed for viremia measurement. At experiment end point 4 dpi (for 28 dpv challenge group) and 7 dpi (for 84 dpv challenge group), ipsilateral and contralateral hind limb tissues (ankles, calves, and quadriceps), spleen, and blood were harvested. Tissues were homogenized in 1 ml of PBS, debris was pelleted and lysates were titered along with sera for the presence of infectious virus on Vero cells by limiting dilution plaque assay. For both short- and long-term vaccine groups, AdV-MAYV vaccination prevented the development of viremia; no infectious MAYV was detectable in serum at 2 dpi while AdV-GFP control animals had a mean viral titer of $8.39x10^4$ and $1.31x10^5$ PFU/ml, respectively ($P < 0.0001$)

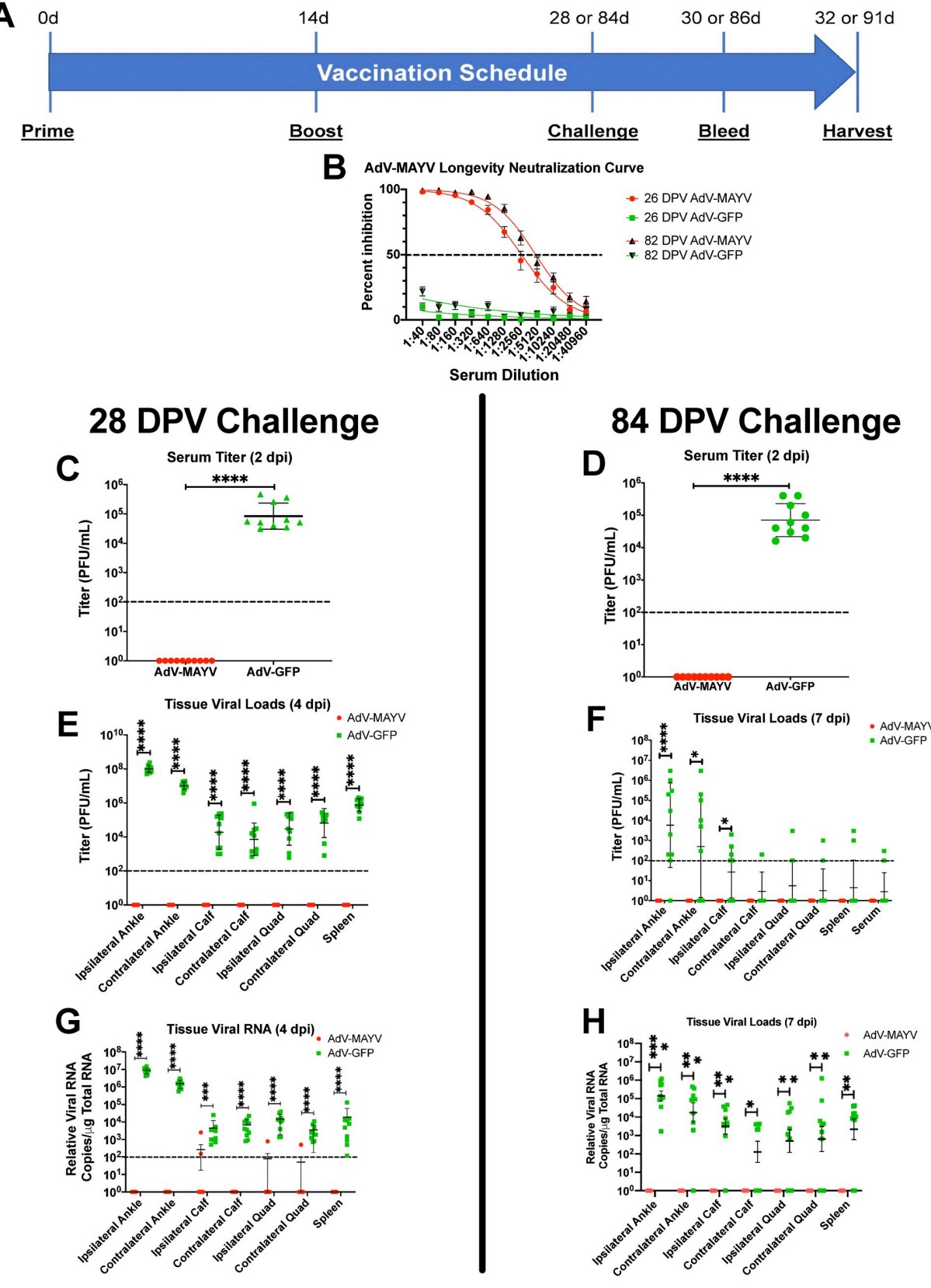

**Fig 4. AdV-MAYV Vaccination Protects WT Mice from MAYV challenge.** (A) WT C57BL/6N mice were vaccinated with a AdV-MAYV or AdV-GFP prime by i.m. injection followed by a booster vaccination at 14 days. At day 28 or 84 post prime, mice were challenged with $10^4$ PFU MAYV$_{BeAr}$ in the right footpad. Blood was collected at 2 dpi and tissues and blood were harvested at 4 or 7 dpi. The data represent a single experiment performed with an n = 10 mice per group. (B) Serum collected prior to challenge displayed robust neutralizing antibody titers for AdV-MAYV vaccinated mice at both 26 and 82 dpv compared to AdV-GFP controls. PRNT50 values calculated for each group by variable slope non-linear regression. Error bars represent SEM. (C and D) Serum viremia at 2 dpi was measured by limiting dilution plaque assay on Vero cells. Viral titers in the serum from AdV-MAYV vaccinated animals was below the detection limit (100 PFU/ml of serum) for all animals. Statistical analysis was performed on log-transformed data using an unpaired Mann-Whitney U test (**** P <0.0001). (E and F) Infectious viral loads in lysates derived from the ankles, calves, quads, spleen tissues and serum were measured by limiting dilution plaque assays at 4 dpi. Infectious viral loads in AdV-MAYV vaccinated animals were below the detection limit for the assay (100 PFU/ml of lysate). Statistical analysis was performed on log-transformed data using unpaired Mann-Whitney U tests (* P < 0.05, **** P < 0.0001). (G and H) Total RNA was extracted from mouse tissue lysates and viral RNA levels were measured by qRT-PCR using primers and probes directed against the virus. Statistical analysis was performed on log-transformed data using unpaired Mann-Whitney U tests (* P < 0.05, ** P < 0.005, *** P = 0.0001, **** P < 0.0001). Black dotted line indicates limit of detection (100 copies per µg of total RNA). Viral RNA was below the detection limit for most AdV-MAYV vaccinated animals following challenge. Error bars in panels C-H represent SD.

(**Fig 4C and 4D**). Similarly, in contrast to control mice, infectious viral loads in all of the tissues tested was below the detectable limit for AdV-MAYV vaccinated mice (P < 0.0001) (**Fig 4E and 4F**). Total RNA was extracted from a portion of the tissue homogenates for qRT-PCR quantification of viral RNA. Control AdV-GFP treated animals all contained viral genomes in each of the surveyed tissues while the majority of animals from the AdV-MAYV vaccinated groups had below detectable levels of viral RNA (P < 0.0001) (**Fig 4G and 4H**). These data indicate that the AdV-MAYV vaccination elicits durable potent neutralizing antibodies that limit viremia and widespread viral tissue distribution.

Next, we determined whether AdV-MAYV vaccination affects the inflammatory immune environment in the joint following MAYV challenge. Tissue homogenates from vaccinated WT mice collected and prepared at 4 dpi were analyzed for cytokine and chemokine levels using a magnetic bead multiplex assay. Both contralateral and ipsilateral ankles from AdV-MAYV vaccinated, MAYV-challenged mice had significantly lower levels of MCP-1, MIP-1α, RANTES, Eotaxin, and MIP-2α when compared to control AdV-GFP vaccinated, MAYV-challenge mice (**Fig 5**). In fact, for all chemokines there was no statistical difference in chemokine levels between AdV-MAYV vaccinated mice challenged with MAYV and the uninfected control mice. The lower levels of inflammatory chemokines following MAYV challenge in AdV-MAYV vaccinated mice correlates with the ability of the vaccination platform to diminish infection. To further evaluate these effects, we challenged groups of WT mice one day after passive transfer of naïve or immune sera from vaccinated mice and compared them to a mock challenged group. At 7 dpi mice in the naïve serum transfer group had significant increases in footpad swelling compared to the immune sera group (P < 0.05) (**Fig 6A**). H&E staining of lower hind limbs harvested from mice at 7 dpi revealed significantly increased pathologic changes in both ipsilateral and contralateral lower leg tissues for the naïve serum group when compared to the MAYV vaccine immune sera group (**Fig 6B**). Representative images from the ankle joint, footpad muscle, and tibia muscle are shown in **S2 Fig**.

Based on the strong protective immunity afforded by AdV-MAYV vaccination to WT mice, we next evaluated vaccine efficacy in IFNαR1$^{-/-}$ mice, which have greatly reduced innate immune responses making them highly susceptible to MAYV infection [46,47]. A group of AdV-MAYV vaccinated IFNαR1$^{-/-}$ mice (n = 7) displayed pre-challenge levels of neutralizing activity against MAYV$_{CH}$ (PRNT$_{50}$ = 1,115) (**Fig 7A**) mirroring the similarly strong neutralizing antibody response identified in the WT mice. At 28 dpv the mice were challenged with $10^4$ PFU MAYV$_{BeAr}$ in the right posterior footpad. At 2 dpi, serum viremia was below detection in AdV-MAYV vaccinated mice in contrast to the viremia level in AdV-GFP vaccinated and PBS control mice ($5.6 \times 10^7$ and $1.2 \times 10^8$ PFU/ml, respectively; P < 0.0001) (**Fig 7B**). Mice were monitored daily for morbidity and mortality until 7 dpi. All AdV-GFP and PBS control mice

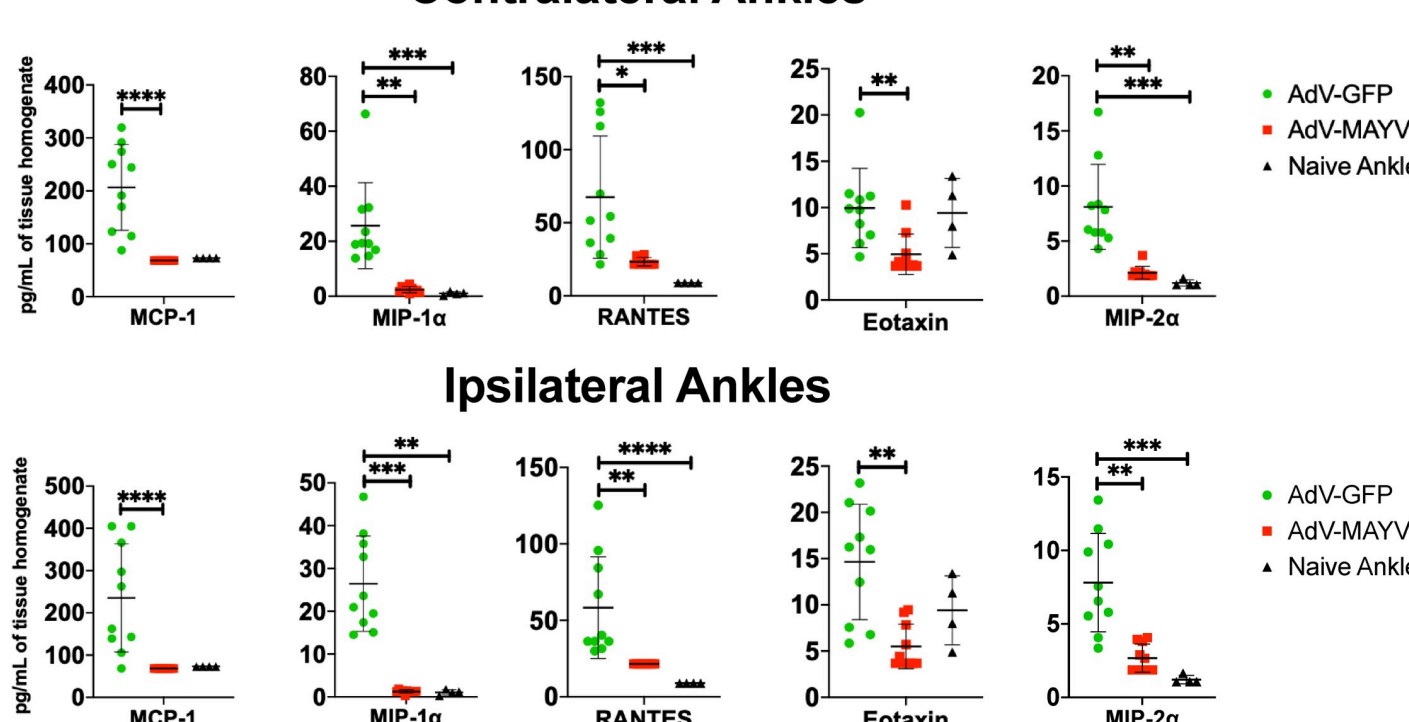

**Fig 5. MAYV-AdV Vaccination Reduces Inflammatory Mediators in the Joint.** Levels of cytokines and chemokines in control mice and mice challenged with MAYV following vaccination with AdV-GFP and AdV-MAYV. WT C57BL/6N mice were vaccinated with AdV-MAYV or AdV-GFP by i.m. injection followed by a booster vaccinated at 14 days. At day 28 post vaccination, mice were challenged with $1x10^4$ PFU/ml MAYV$_{BeAr}$ in the right footpad (n = 10 mice per group). Ankle tissue homogenates collected from mice at 4 dpi were analyzed for cytokine and chemokines by a 26-plex cytokine multiplex kit and compared to naïve tissues. Statistical analysis was performed using Kruskal-Wallis tests and error bars represent SD (* $P < 0.05$, ** $P < 0.005$, *** $P = 0.0001$, **** $P < 0.0001$).

succumbed to infection by 5 dpi, while AdV-MAYV mice survived without physical signs of infection until the study endpoint at 7 dpi (P < 0.0001) (**Fig 7C**). The AdV-MAYV vaccine elicited similar levels of neutralizing antibodies in both WT mice and IFNαR1$^{-/-}$ mice, which afforded them with protection against lethal challenge.

### Passive transfer of AdV-MAYV vaccinated serum provides protective immunity

To assess the ability of circulating antibodies to protect IFNαR1$^{-/-}$ mice against MAYV infection, sera collected from AdV-MAYV vaccinated WT mice at 28 dpv was i.p. injected into IFNαR$^{-/-}$ mice one day prior to lethal challenge with $10^4$ PFU MAYV$_{BeAr}$ and mice were monitored daily for signs of infection and survival for 7 days (Experimental design is shown in **Fig 8A**). Blood collected at 2 dpi was used to measure differences in serum viremia levels using plaque assays. IFNαR1$^{-/-}$ mice receiving serum from AdV-MAYV vaccinated mice had significantly reduced levels of infectious virus compared to IFNαR1$^{-/-}$ mice receiving serum from PBS control mice (P = 0.008) (**Fig 8B**). In concordance with reductions in levels of serum viremia, IFNαR1$^{-/-}$ mice receiving serum from AdV-MAYV WT mice maintained their starting body weights in contrast to the PBS control group, which lost 14% from their starting weights the day prior to death (P << 0.0001) (**Fig 8C**). As shown in **Fig 8D**, control serum transfer IFNαR1$^{-/-}$ mice survived only to 4 dpi while IFNαR1$^{-/-}$ mice receiving serum from AdV-MAYV vaccinated WT mice all survived until study endpoint at 7 dpi (P < 0.0001).

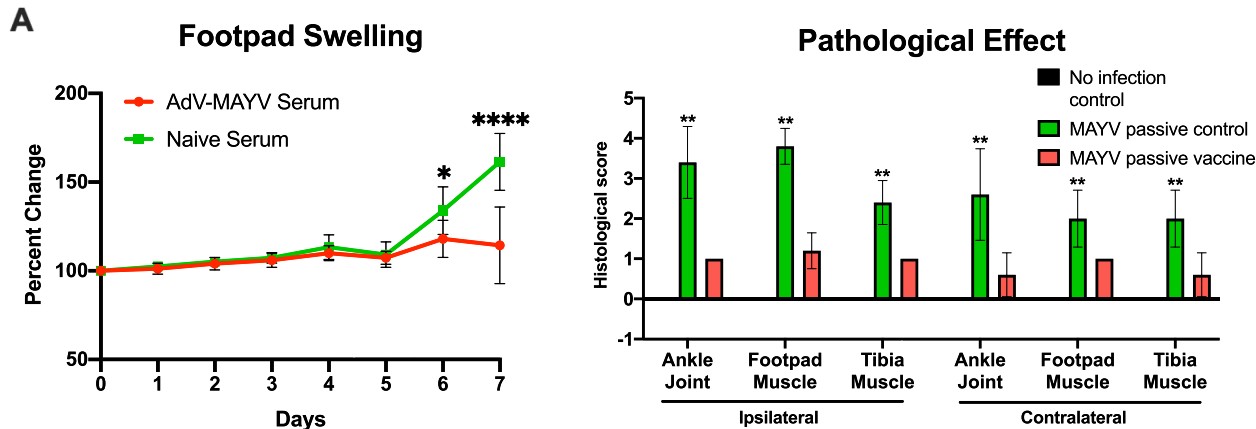

**Fig 6. Passive Transfer of AdV-MAYV Immune Sera Protects Against Pathological Effects of Infection.** Naïve or immune sera was passively transferred to groups of five-week-old female WT C57BL/6N mice one day prior to challenge with $10^4$ PFU MAYV$_{BeAr}$ in the right hind limb footpad. A second group of naïve five-week-old female WT C57BL/6N were mock inoculated with PBS. (A) Footpad swelling was monitored by digital calipers throughout the experiment. Statistical analysis was performed by paired repeated measures ANOVA (* P < 0.05, **** P < 0.0001). (B) Whole hind legs were harvested at 7 dpi and sectioned for histopathology by hematoxylin and eosin staining. Tissue sections were scored on a 0–5 scale using the no infection control tissues as a baseline score: 0 absent (no lesions), 1 minimal (1~10% of tissues affected), 2 mild (11~25% affected), 3 moderate (26~50% affected), 4 marked (51~75% affected), 5 severe (>75% affected). Statistical analysis was performed by two-way ANOVA (* P <0.05, **** P < 0.0001). Error bars represent SD.

Tissue viral RNA levels at 7 dpi in IFNαR1$^{-/-}$ mice receiving serum from AdV-MAYV vaccinated WT mice were detectable only in the ipsilateral ankle, except for one mouse with viral RNA detected in the contralateral ankle (Fig 8E). These data demonstrate the protective nature of the strongly neutralizing antibodies in the serum of AdV-MAYV vaccinated mice, but indicate the fact they are not the sole mediators in viral protection.

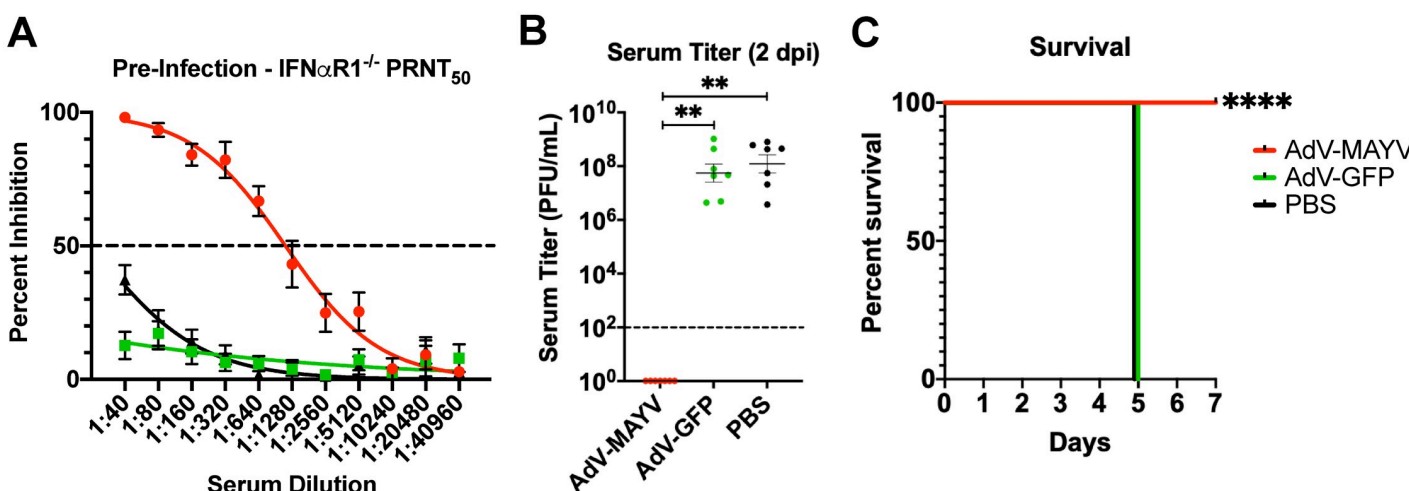

**Fig 7. AdV-MAYV Vaccination Protects IFNαR1$^{-/-}$ Mice from Lethal Challenge.** Male and female IFNαR1$^{-/-}$ mice were vaccinated with $10^8$ PFU AdV-MAYV, AdV-GFP, or PBS by i.m. injection followed by a booster vaccination 14 days later. At day 28, mice were challenged with $1\times10^4$ PFU/ml MAYV$_{BeAr}$ in the right footpad. The data presented represents one independent experiment (n = 7 mice per vaccine). (A) Prior to challenge, blood was collected to measure neutralizing titers against MAYV$_{CH}$ by PRNT assay. AdV-MAYV elicited robust neutralizing antibodies in IFNαR1$^{-/-}$ mice. Error bars represent SEM. (B) Limiting dilution plaque assays were used to measure viremia for blood serum samples collected at 2 dpi. Black dotted line indicates limit of detection (100 PFU/ml). Statistical analysis was performed on log-transformed data using a Kruskal-Wallis test (** P < 0.005). Error bars represent SD. (C) Mouse morbidity and mortality was monitored daily for 7 days post infection with Kaplan-Meier survival curve analysis (**** P < 0.0001).

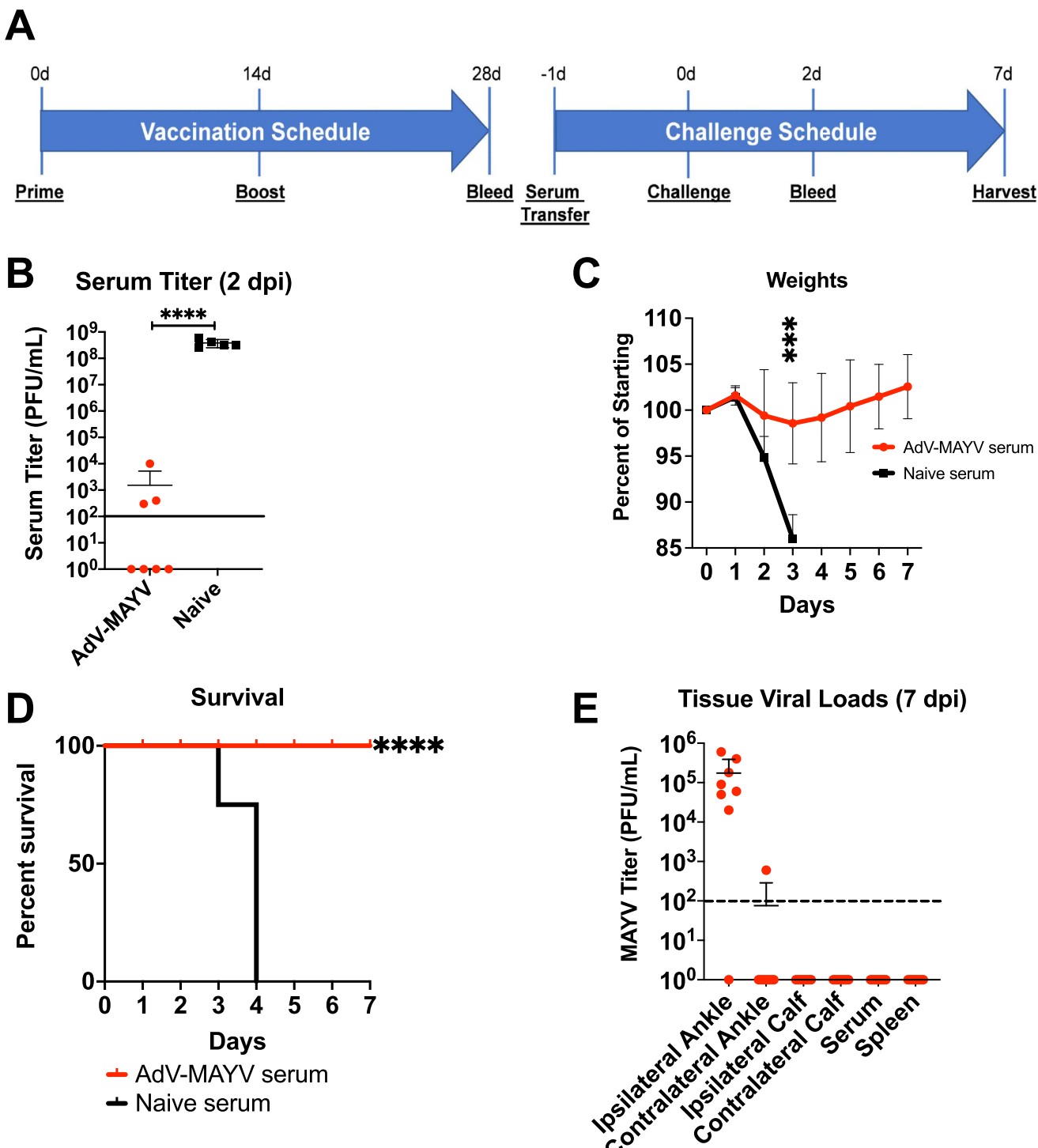

**Fig 8. Passive Transfer of Immune Serum Protects IFNαR1$^{-/-}$ Mice from Lethal MAYV Challenge.** (A) Six WT C57BL/6N mice were vaccinated with $10^8$ PFU of AdV-MAYV following the prime + boost regimen; at 28 days post-prime total blood was collected and serum pooled from all mice. A bolus of 200 μl of pooled serum from AdV-MAYV vaccinated or naive mice was administered to IFNαR1$^{-/-}$ mice by intraperitoneal injection 1 day before challenge with $1 \times 10^4$ PFU MAYV$_{BeAr}$. The data represents one experiment with n = 7 per condition. (B) Blood collected at 2 days post-challenge was used to measure viremia by limiting dilution plaque assays. While all 7 animals receiving control serum had high levels of virus, only three of the seven animals receiving passive transfer of immune sera had detectable virus, which was 5–6 logs lower than controls. Statistical analysis was performed on log-transformed data using a Mann-Whitney test ($^{**}$ P < 0.005). (C) Mice were weighed daily after challenge until experiment endpoint at 7 dpi. Statistical analysis was performed using

multiple repeated measures mixed-effects ANOVA (*** P = 0.0001). (D) Mouse survival following MAYV challenge was graphed. Statistical analysis was performed using Kaplan-Meier survival curve analysis (**** P < 0.0001). (E) Tissue viral RNA levels were determined by qRT-PCR for mice that survived until 7 dpi (those animals receiving AdV-MAYV vaccine sera only). Virus was detected in the ipsilateral ankles of challenged mice but very little was detected in other tissues. Black dotted line indicates limit of detection (100 viral RNA copies/μg of total RNA). Error bars represent SD.

### AdV-MAYV vaccination elicits cross-protection against other alphaviruses

Previous studies have indicated the existence of cross-protection for related alphaviruses such as chikungunya virus (CHIKV) and MAYV [29,30]. Thus, we next evaluated the AdV-MAYV vaccine for protective immunity against CHIKV and UNAV, two related alphaviruses in the Semliki Forest complex [37]. Serum collected from WT mice vaccinated with AdV-MAYV was first tested for neutralizing activity against CHIKV and UNAV and shown by PRNT to reduce the levels of infection for both viruses (**Fig 9A and 9B**). Based on this data, *in vivo* cross-protection experiments in lethally challenged IFNαR1$^{-/-}$ mice were conducted to evaluate the protective efficacy of AdV-MAYV immunization against CHIKV or UNAV infection. IFNαR1$^{-/-}$ mice were vaccinated and challenged at 28 dpv with either $10^3$ PFU CHIKV$_{SL15649}$ or $10^4$ PFU UNAV$_{Mac150}$ in the right footpad. Blood collected at 2 dpi from control mice showed high levels of CHIKV and UNAV viremia but viremia in the AdV-MAYV vaccinated groups were below the level of detection (P < 0.0001 for both groups) (**Fig 10A**). Animals were monitored daily until 7 dpi for the presence of ipsilateral footpad swelling, weight loss and other signs of morbidity. All of the control vaccinated animals developed weight loss and clear signs of morbidity; these animals were euthanized by 4 or 5 dpi. AdV-MAYV vaccinated animals did not display weight loss and survived until the study endpoint at 7 dpi (P < 0.0001) (**Fig 10B and 10C**). One mouse in the AdV-MAYV CHIKV challenge group died of reasons unrelated to infection. Control animals challenged with CHIKV had remarkable footpad swelling starting at 2 dpi until death at 5 dpi (**Fig 10D**). CHIKV disease, weight loss, and associated death in IFNαR1$^{-/-}$ mice was completely abrogated by AdV-MAYV vaccination (**Fig 10B–10D**). There was no detectable footpad swelling in mice challenged with UNAV but this might take longer to develop than for CHIKV challenge (**Fig 10C**). Tissues and sera were collected from the surviving AdV-MAYV vaccinated IFNαR1$^{-/-}$ mice at 7 dpi to measure infectious virus and viral genomes (**Fig 10E–10G**). For both CHIKV and UNAV challenged mice the ipsilateral ankle was the only tissue with detectable infectious virus with 4 of 5 mice in the CHIKV challenge and 1 out of 6 mice in the UNAV challenge groups having viral titers above the limit of detection (**Fig 10E**). We predict that the low levels of virus present in the ipsilateral ankle are not sufficient to promote footpad swelling in the CHIKV challenge group. CHIKV or UNAV vRNA RT-qPCR analysis of tissues identified the presence of low levels of viral genomes in the ankle and calf tissues of surviving vaccinated challenge group mice. UNAV challenged IFNαR1$^{-/-}$ mice had equivalent levels of viral genomes on both contralateral and ipsilateral tissues, with ankle tissues having a mean approximately 2 logs higher than calf tissues (**Fig 10F**). In CHIKV challenged IFNαR1$^{-/-}$ mice there was a notable difference in viral genomes based upon proximity to infection site. The ipsilateral ankle had a mean approximately 3.2 logs higher than the contralateral ankle, and while viral genomes were detected in the ipsilateral calf, no genomes were detected on the contralateral calf (**Fig 10G**). These findings replicate the previous results in the vaccinated MAYV challenged mice and indicate that the vaccine platform generates cross-protective immunity against related alphaviruses.

### Discussion

In this report, we demonstrate that a non-replicating human adenovirus serotype 5 based vaccine platform expressing the MAYV structural protein (AdV-MAYV) generated potent

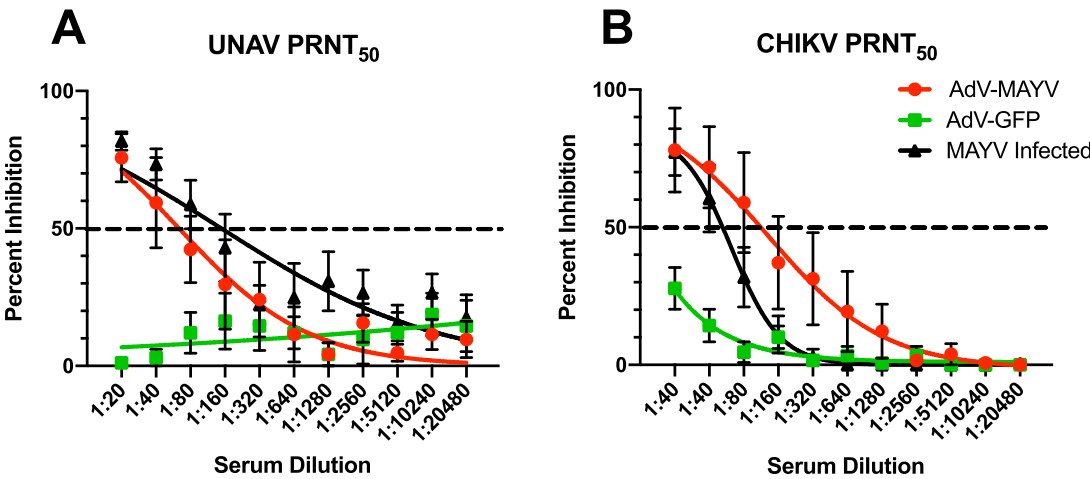

**Fig 9. AdV-MAYV Elicits Cross Neutralizing Antibodies Against Una and Chikungunya Viruses.** In order to determine whether the AdV-MAYV elicited cross-protection against related alphaviruses, PRNT$_{50}$ assays were performed on serum from WT C57BL/6N mice vaccinated with AdV-MAYV, AdV-GFP vaccinated, and compared to serum from mice infected with MAYV. Shown are the average PRNT$_{50}$ values calculated for each group (n = 5). PRNT50 assays for (A) Una virus and (B) CHIKV indicate cross-species neutralization is elicited in serum from mice vaccinated with AdV-MAYV as well as mice infected with MAYV but not for serum collected from AdV-GFP vaccinated controls. Curves were calculated using a variable slope non-linear regression analysis. Error bars representing SEM from 5 biological replicates.

immunogenicity. AdV-MAYV generated virus-like particles that were shed from transduced cells, which may be an important attribute that leads to robust neutralizing antibody generation. As such, in a mouse MAYV challenge model, AdV-MAYV vaccination resulted in a significant decrease in viremia at 2 dpi to levels that were below the detection limit when compared to AdV-GFP vaccine control mice. This finding was confirmed during testing for infectious virus and viral genomes in tissues at 4 dpi. Upon challenge, IFNαR1$^{-/-}$ mice vaccinated with AdV-MAYV also had undetectable levels of viremia at 2 dpi and all survived to the study endpoint at 7 dpi. This demonstrated that even in a very stringent challenge model that the vaccine elicited highly efficacious immunity. Mechanism of protection studies identified AdV-MAYV vaccination elicited highly neutralizing and binding MAYV specific antibodies as well as E2 protein specific T cell responses. Epitopes in E2 have been identified as prominent targets of early T cell responses in CHIKV infection and support the N-terminus of E2 as an important target of the early adaptive immune response [39,48]. Interestingly, the AdV-MAYV vaccine also protected IFNαR1$^{-/-}$ mice from challenge with other Old-world alphaviruses including UNAV or CHIKV indicating cross protection elicited by the vaccine.

IFNαR$^{-/-}$ mice are a highly stringent model as they display increased virus replication and tissue damage compared to wildtype mice [46]. Previous studies have indicated a high susceptibility of IFNαR1$^{-/-}$ mice to members of the alphavirus family with lethality achieved using doses of as little as 3 PFU CHIKV or approximately $10^2$–$10^4$ PFU when challenged with other family members including ONNV and VEEV [46,47]. Thus, they are highly effective models to understand the protective effects of vaccination approaches [46,49,50]. Passive transfer experiments demonstrated robust and transferrable antibody production in AdV-MAYV vaccinated mice, partially protecting recipient mice from viremia at 2 dpi and enabling survival until study endpoint. Also observed was the ability to largely reduce or entirely ablate the presence of infectious virus and/or viral RNA in the muscle and joint tissues of vaccinated and challenged mice. Joint and muscle tissues are known to be targets for arthritogenic alphavirus

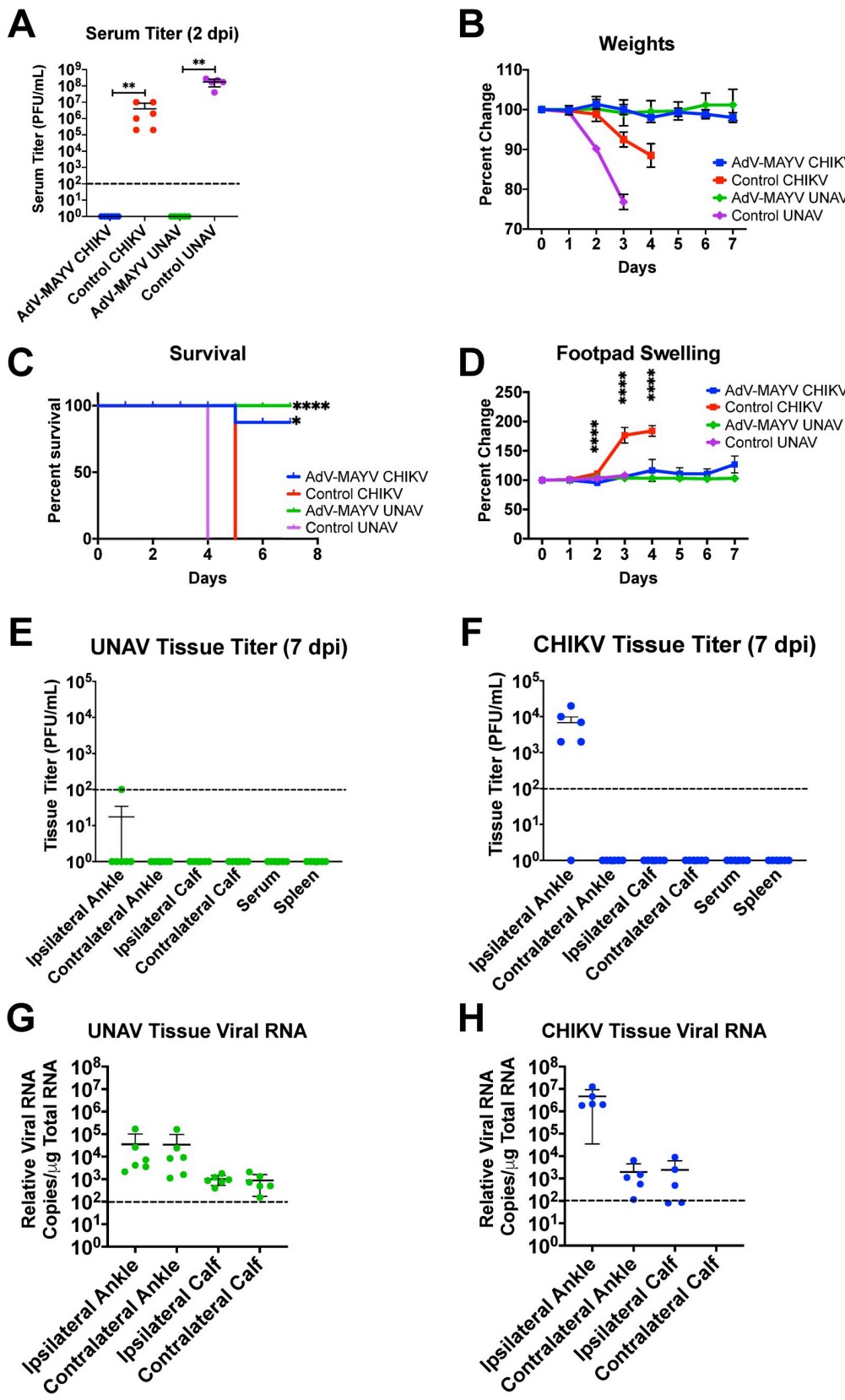

**Fig 10. AdV-MAYV Vaccination Cross-Protects IFNAR1$^{-/-}$ Mice Against Lethal Challenge with CHIKV and Una Virus.** IFNαR1$^{-/-}$ mice vaccinated by intramuscular injection of AdV-MAYV, using the prime-boost vaccination regimen, were challenged with $1 \times 10^4$ PFU CHIKV or UNAV in the right footpad. Serum was collected at 2 dpi and animals were monitored for clinical signs for 7 days. Experimental timeline is similar to that in Fig 4A. Data represents one independent experiment performed with n = 5 mice per group for vaccinated animals and n = 6 for controls. (A) Viremia was measured by limiting dilution plaque assay on confluent Vero cells. Statistical analysis was performed within groups by Mann-Whitney tests (** P < 0.005). Following challenge mice were monitored daily until experiment endpoint for (B) weight loss (C) morbidity and mortality (D) changes in right posterior footpad swelling as measure by caliper. Statistical analysis was performed using multiple T-tests and Kaplan-Meier survival curve analysis. Error bars represent SD (* P < 0.05, ** P < 0.005, **** P < 0.0001). At 7 dpi tissues were harvested and ankle, calf, and spleen infectious viral loads for Una virus (E) and CHIKV virus (F) were measured in tissue homogenates by limiting dilution plaque assays. Black dotted line indicates limit of detection (100 copies of viral RNA/ μg of total RNA). Total RNA was extracted from ankle and calf tissue homogenates to quantify Una virus (G) and CHIKV virus (H) RNA loads by qRT-PCR. Error bars represent SD.

replication. Fibroblasts, mesenchymal, and osteoblast cells have been identified as predominant and preferential locations for CHIKV replication, and muscle satellite cells have also been identified as selective targets in human muscle tissues [51,52]. Infection of joints typically leads to arthralgia in small joints (e.g. fingers, wrist, tarsus) prior to larger joints (e.g. knees and shoulders) but can and typically does involve multiple joints simultaneously [2]. High levels of viral replication and persistence in these tissues as well as immune cell infiltration and long-term inflammation are both responsible for the prolonged arthralgia and myalgia experienced by patients. We demonstrate that the AdV-MAYV vaccine can play an important role in diminishing the inflammatory disease experienced following infection by restricting viral dissemination and replication in these tissues and thus limiting the associated inflammatory response. It has previously been shown that adult IFNαR$^{-/-}$ mice challenged with MAYV$_{TR4675}$ exhibit high levels of infectious virus in the footpad, knee, gastrocnemius, and thigh muscle, while adult C57BL/6N had high levels in their thigh, spleen, and gastrocnemius [53]. Our findings agree with these data and also suggest the spleen is a target of viral infection permissive to high levels of MAYV replication in the absence of innate immunity (**Fig 4**). Further, while vaccination was able to prevent viral dissemination, passive transfer of immune serum was capable of significantly reducing the pathological symptoms of infection when we surveyed lower hind limb ankle joints, and footpad and tibia muscles (**Figs 6 and S2**).

A previous study by *Webb et. al.*, in which mice vaccinated with a CHIKV/IRES vector, as well as CHIKV infected mice, produced limited MAYV cross-reactive antibodies, indicating the existance of shared antigenic epitopes between MAYV and CHIKV [32]. That study did, however, indicate that a minimum level of CHIKV neutralizing antibody titer must be reached in order to confer cross-protection, supporting the idea that robust stimulation of neutralizing antibodies is key for broad protective effects. Our AdV-MAYV vaccination regimen also supports these findings and provides confirmatory evidence that shared antigenic domains exist between these related alphavirus member species. Upon testing serum from vaccinated mice, it was determined that the elicited antibodies bore close similarity to antibody types and levels produced in MAYV infected mice, although total Ig was significantly decreased between MAYV infected and prime + boosted AdV-MAYV vaccinated mice (**Fig 3**). IgG3 was the only subclass induced by prime/boost vaccination that displayed higher binding levels than those responses in the serum from infected mice. It is the only IgG subclass that is T cell independent and primarily is targeted against carbohydrates and repeating protein aggregates [54–56]. Studies have previously indicated that IgG3 subclass antibodies were the predominant anti-E2 glycoprotein response in mice vaccinated with CHIKV VLP and appeared shortly after vaccination [57]. IgG2b antibodies work with IgG3 in the early T cell independent response but function to bring early FcγR-mediated effector functions [56]. This partnership could explain

why even if IgG3 primarily binds to E2 epitopes the expression is diminished at 7 dpi while IgG2b is elevated in order to direct ADCC and CDC functional responses. The strong levels of IgG1 and IgG2b are also in line with previous studies that utilized AdV vectors in 129Sv/Ev and IFNαR$^{-/-}$ mice where significant production of these subclasses was noted [58]. The ability to stimulate both strong humoral and cellular immune responses correlates with evidence presented by Choi *et. al.* in their DNA based vaccine against MAYV [8,10,59]. Surveying mouse ankles identified that there were elevated levels of MCP-1, MIP-1α, RANTES, Eotaxin, and MIP-2α. Recently, the analysis of spleens from MAYV/IRES vaccinated and challenged mice had a significant reduction in MCP-1, MIP-1α, and RANTES, among others, compared to MAYV infected at 3 dpi [11]. We previously identified these among a group of elevated chemokines in sampled ipsilateral ankle tissues in control mice following infection with CHIKV, and others have found that MCP-1 and MIP-1α were both elevated in tissues of Ross River infected mice by qRT-PCR [39,60]. Thus, the reduction in these inflammatory chemokines could be important factors in reducing the sequalae of inflammatory responses in infected individuals. Our balanced cellular and humoral immune response elicited by the AdV-MAYV provides robust neutralizing antibodies but also T cell responses that we have previously shown are important for limiting inflammation following alphavirus challenge [39].

Adenovirus based vaccine vectors have advantages when compared to live-attenuated and recombinant protein vaccines with their ability to stimulate both strong humoral and cellular immune responses, elicit strong persistent immunity, and the ability to be used in susceptible populations such as the elderly and immunocompromised [25,61–63]. They can be readily produced to high titer, do not integrate, and can be administered safely. Our AdV-MAYV vaccine approach against MAYV has ability to cross-protect against both CHIKV and UNAV, which is distinct from other reported MAYV vaccines, although it could very well be that these other vaccines did not address this feature [8,10,12,32,59]. Our results agree with the recent publication by Kroon Campos *et al.* where they showed that mice vaccinated with ChAdOx1 vectors expressing Capsid, Envelope 1–3 and 6K structural proteins were afforded full protection against MAYV challenge and partial protection against heterotypic challenge with CHIKV [12]. Reciprocal findings were observed when mice were vaccinated with a ChAdOx1 vector expressing the Capsid, Envelope 1–3 and 6K structural proteins from CHIKV. Differences in study designs exist, most notably vaccine regimen, structural protein cassettes, and vector serotype. As has been indicated by prior studies, a single dose regimen as used by Kroon Campos *et al.* may have been insufficient to obtain neutralizing antibody titers capable of robust protection against heterotypic viruses [32]. Similarly, as TF has previously been indicated as predominantly incorporated into assembled virions rather than 6K, it's absence may have altered the production and secretion of true-to-form VLP [64]. Differences between vector serotype may have also affected the immunogenicity of our vaccine vectors and effected the elicitation of humoral and cellular immune responses as has previously been noted between human and chimpanzee serotypes [65–67]. CHIKV and UNAV circulate in the same geographical region as MAYV, so providing multitarget protection would be beneficial to local inhabitants and travelers alike. There have been no previous studies on the development of UNAV vaccines, nor the indication that previously published pre-clinical alphavirus vaccines provide cross protection against UNAV, thus making the AdV-MAYV novel in this regard. Previous studies in macaques reported on the ability of antibodies produced during CHIKV infection to neutralize MAYV, and serum samples from humans indicated that convalescent CHIKV infected patient antibodies were capable of neutralizing both MAYV and UNAV support the data presented [30–32]. Similar vaccine studies in mice have also observed heterotypic protection against related alphaviruses, indicating that this phenomenon is a feature that should be studied alphavirus vaccine development [33–36]. This data confirms the findings

that shared epitopes for neutralizing antibodies exist between these members of the Semliki Forest complex and supports the presented findings of cross protection from our AdV-MAYV vaccine. Additional pre-clinical studies with this vaccine vector will provide important insight into new approaches to vaccinate at risk populations against MAYV, Una, and CHIKV.

## Supporting information

**S1 Fig. AdV-MAYV Vaccination Elicits Robust T-cell Response Against MAYV E2 Glycoprotein in Wild Type and IFNαR1$^{-/-}$ Mice.** C57BL/6N or IFNAR$^{-/-}$ mice were vaccinated with AdV-MAYV or AdV-GFP by i.m. injection followed by a booster vaccination 14 days later. At day 28 post vaccination, spleens were collected and processed for lymphocytes. IFNγ ELISpot assays were performed by stimulating $2.5 \times 10^5$ splenocytes with 18mer peptides from the MAYV structural proteins incorporated, DMSO (vehicle negative control) or PMA/ionomycin (positive control). At 2 days post stimulation the plates were developed for the presence of IFNγ and spots were counted using an automated microscope with computer interface. Two independent experiments were performed with 4 biological replicates. Statistical analysis was performed by paired one-way ANOVA and error bars represent SD (* $P < 0.05$, ** $P < 0.01$).
(EPS)

**S2 Fig. Histological Analysis Demonstrates Protective Effects of AdV-MAYV Vaccine Immune Sera Delivered by Passive Transfer to Wild Type Mice.** Two groups of C57BL/6N mice received passive transfer of serum from naïve mice (Control Vaccine, middle row) or AdV-MAYV vaccinated mice (MAYV vaccine, bottom row) and were infected with $10^4$ PFU MAYV$_{BeAr}$ in their right hind footpad. A group of naïve wildtype controls (uninfected, top row) were mock challenged with PBS. At 7 dpi mice were sacrificed and perfused with 4% paraformaldehyde in PBS. Lower hind limbs were harvested, decalcified, embedded in paraffin, and 5 μm sections were prepared for H&E analysis. Shown are representative images of gross pathology for the ankle joint, footpad muscle, and tibia muscle between the three groups. Magnification was 40x or 100x as indicated.
(EPS)

## Acknowledgments

Electron microscopy was performed at the Multiscale Microscopy Core with technical support from the Oregon Health & Science University-FEI Living Lab and the OHSU Center for Spatial Systems Biomedicine.

## Author Contributions

**Conceptualization:** John M. Powers, Nicole N. Haese, Rebecca Broeckel, Daniel N. Streblow.

**Data curation:** John M. Powers, Nicole N. Haese.

**Formal analysis:** John M. Powers, Nicole N. Haese, Daniel N. Streblow.

**Funding acquisition:** Mark T. Heise, Daniel N. Streblow.

**Investigation:** John M. Powers, Nicole N. Haese, Michael Denton, Takeshi Ando, Craig Kreklywich, Kiley Bonin, Cassilyn E. Streblow, Nicholas Kreklywich, Patricia Smith, Rebecca Broeckel.

**Methodology:** Thomas E. Morrison, Mark T. Heise, Daniel N. Streblow.

**Project administration:** Daniel N. Streblow.

**Resources:** Victor DeFilippis, Mark T. Heise, Daniel N. Streblow.

**Supervision:** Mark T. Heise, Daniel N. Streblow.

**Writing – original draft:** John M. Powers, Nicole N. Haese, Daniel N. Streblow.

**Writing – review & editing:** John M. Powers, Nicole N. Haese, Rebecca Broeckel, Victor DeFilippis, Thomas E. Morrison, Mark T. Heise, Daniel N. Streblow.

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
