## [Decision Letter · Decision Letter 0]

4 Dec 2020

Dear Dr. Streblow,

Thank you very much for submitting your manuscript "Non-Replicating Adenovirus Based Mayaro Virus Vaccine Elicits Protective Immune Responses and Cross Protects Against Other Alphaviruses" for consideration at PLOS Neglected Tropical Diseases. As with all papers reviewed by the journal, your manuscript was reviewed by members of the editorial board and by several independent reviewers. In light of the reviews (below this email), we would like to invite the resubmission of a significantly-revised version that takes into account the reviewers' comments. 

We cannot make any decision about publication until we have seen the revised manuscript and your response to the reviewers' comments. Your revised manuscript is also likely to be sent to reviewers for further evaluation.

Sincerely,

Ran Wang, Ph.D., M.D.

Associate Editor

Francis Jiggins

Deputy Editor

Reviewer's Responses to Questions

**Key Review Criteria Required for Acceptance?**

**Methods**

-Are the objectives of the study clearly articulated with a clear testable hypothesis stated?

-Is the study design appropriate to address the stated objectives?

-Is the population clearly described and appropriate for the hypothesis being tested?

-Is the sample size sufficient to ensure adequate power to address the hypothesis being tested?

-Were correct statistical analysis used to support conclusions?

-Are there concerns about ethical or regulatory requirements being met?

Reviewer #1: The methods are adequately chosen and described in detail. The conclusions drawn by the authors are well supported be the data.

The animal experiments met regulators requirements.

Reviewer #2: (No Response)

Reviewer #3: -Are the objectives of the study clearly articulated with a clear testable hypothesis stated? - yes

-Is the study design appropriate to address the stated objectives? - yes

-Is the population clearly described and appropriate for the hypothesis being tested? - Not applicable

-Is the sample size sufficient to ensure adequate power to address the hypothesis being tested? - yes

-Were correct statistical analysis used to support conclusions? - yes

-Are there concerns about ethical or regulatory requirements being met? - no

Reviewer #4: -Are the objectives of the study clearly articulated with a clear testable hypothesis stated?

Yes

-Is the study design appropriate to address the stated objectives?

Yes

-Is the population clearly described and appropriate for the hypothesis being tested?

N/A

-Is the sample size sufficient to ensure adequate power to address the hypothesis being tested?

Yes

-Were correct statistical analysis used to support conclusions?

Yes

-Are there concerns about ethical or regulatory requirements being met?

Yes

Materials and methods comments

Viruses

Why do you have to add both High and Low viscosity CMC on infected cell monolayer?

Adenovirus Vaccine Vector

For amplification of MAYV gene, before the PCR product amplification, you need the cDNA production, since you are working with an RNA virus,. Please include this comment

Mouse experiments

1. For viral feet challenge, why did you use 10-log reduced viral load for CHIKV (10^3 PFU) compared to MAYV and UNAV?

Quantification of virus tissue burden

1. It is mentioned that you used 500uL of PBS for tissue harvest. If you used the same PBS volume for all collected tissue samples, I assume they would present different tissue concentrations, depending on sample size/mass. How did you normalized your PFU/qRT-PCR and cytokines results? I see that your tissue viral loads (PFU/mL and copies/mL) and cytokines quantification (pg/mL) are all in mL of tissue homogenate.

2. Which frequency and equipment did you use for tissue homogenization? Please include this information

3. '0.2% methyl blue': In Viruses section it is described '0.5% methylene blue'. Please, double-check

qRT-PCR

1. The MAYV reverse primer does not anneal in a reverse way to the target region. The complementary reverse sequence would be '5- CTTCCAGGCTGCCCGGCACCAT -3'. Please, double-check the sequence;

Western Blot Analysis

1. "Cells were harvested 72 hpi after washing with PBS and were pelleted at 9,391 x g for 15 minutes at 4C." At this centrifugal g force I would expect a degree of cell lysing. Please, double-check

Neutralization Assays

1. In the case of this experiment, the ideal PRNT positive control would be sera from MOCK infected mice instead of virus only. This sera may have unspecific or natural antibodies that may interfere in the viral neutralization and should better work as a background control.

Enzyme-Linked Immunoassay (ELISA)

1. ELISA coating is typical performed with a basic buffer (pH > 9) such as Carbonate-bicarbonate buffer. Please, double check which buffer did you use and insert a trustful reference of ELISA with total Alphavirus antigens: 10.1371/journal.pntd.0000753

2. Did you washed your plates after viral adsorption? Include this information;

3. How much volume did you use for blocking? Include this information;

4. Did you evaluated your samples in duplicate, triplicate wells...? Include this information;

5. What is the composition of 'ELISA wash buffer' and 'Elisa buffer' ?;

6. Please, give more information about the secondary antibodies used in these experiments;

Enzyme-Linked Immunospot Assay (ELISpot)

1. Did you not sliced, macerated and digested the spleen before cell strainer? It is quite a hard tissue to simple pass through the mesh. Please, double-check;

2. Is there any specific reason to use 2ul volume of DMSO as the control? I suppose it should be the same volume used for the peptides;

3. Please include the ELISpot plate reader model and manufacturer; 

Histopathology

1. You cannot properly cut 5-um sections of tissues that contains bone and cartilages. Did you decalcified and dehydrated the tissue? Removed the skin? Please, include a better description of this methodology and include references.

2. What do you mean by 'severity of histological lesions'? Infiltrates? Necrosis? Hemorrhage? Tissue or cartilage destruction? This point should be clarified for better understanding.

Reviewer #5: see attached documentation

**Results**

-Does the analysis presented match the analysis plan?

-Are the results clearly and completely presented?

-Are the figures (Tables, Images) of sufficient quality for clarity?

Reviewer #1: The experiment are presented in a logic order and very well supported the conclusions the author stated. Figures need minor modifications.

1. Figure 3B. Why are different bands detected with mouse serum compared to CHIKV Abs?

2. Figure 5: The authors should comment on the Eotaxin values. Apparently, AdV-MAYV vaccination inhibits eotaxin compared to the untreated control.

3. Figure 6B: No infection is indicated in the figure legend, but the columns are not visible.

Reviewer #2: (No Response)

Reviewer #3: -Does the analysis presented match the analysis plan? - yes

-Are the results clearly and completely presented? - yes

-Are the figures (Tables, Images) of sufficient quality for clarity? - yes

Reviewer #4: -Does the analysis presented match the analysis plan?

Yes

-Are the results clearly and completely presented?

Yes

-Are the figures (Tables, Images) of sufficient quality for clarity?

Yes, however, the figures are not in good quality

Results comments

Adenovirus Mediated Expression of Mayaro Structural Proteins

1. What is the Anti-alphavirus capsid antibody? Please describe in Methods or in the figure legend, if it is a commercial poly/monoclonal antibody or an in house developed antibody.

2. In the VLP electronic microscopy (Fig 1C), do you have another evidence that those viral particles are the Alphavirus VLPs and not the Adenoviruses?

AdV-MAYV Vaccination Elicits Neutralizing Antibodies and T cell Responses

1. Fig 2C: How do you explain the neutralization activity of Control Serum group? Please comment;

2. In Fig 3B, panel 4, we can see a different pattern of proteins recognition from vaccinated mice that differs from monoclonal antibodies but present some comparable molecular weight. I guess it would be careful only to assume that the antibodies from vaccinated mice probable are recognizing Envelope proteins considering the molecular weight size. Additionally, what evidence do you have that the lower band in Panel 4 is the capsid protein?;

3. Please, give more information about the monoclonal antibodies (87H1 and 133.B4) in Methods, westblotting section;

AdV-MAYV Vaccine Elicits Protective Efficacy & Reduces Inflammatory Chemokine Production

1. Is the Supplementary figure 3 basically showing the same data of Figure 7A? Why do you consider necessary to additionally show it?

2. Furthermore, in the experiment from Supplementary Fig 3 there was a pronounced higher neutralization than animals from AdV-MAYV from Figure 7A, which additionally received a AdV-MAYV boost. What is the reason of that?

Passive Transfer of AdV-MAYV Vaccinated Serum Provides Protective Immunity

1. In the conclusion of this section, I would suggests to include that the results also demonstrate that antibodies are not the single mediators of viral protection, since you have a slight body weight reduction and detectable viremia development in serum (2dpi) and Ankle at (7dpi) in mice that received the sera;

Discussion Section

1. "MAYV infection causes a febrile illness for 3-5 days, while joint and muscle pain can last for up to a year. Based on the similarity of disease symptoms between Old-world alphaviruses and serological cross-reactivity, infections with MAYV are commonly misdiagnosed [3]." This phrase is almost the same as in the Introduction;

2. "Passive transfer experiments demonstrated robust and transferrable antibody production in AdV-MAYV vaccinated mice, protecting recipient mice from viremia at 2 dpi and enabling survival until study endpoint". The passive sera transfer experiment relate to virema detection was performed on IFNAR1-/- mice, which demonstrated reduced viremia when compared to non-vaccinated mice. Thus, I think this phrase should be "Partially protected from virema at 2 dpi".

3. "Our findings agree with these data and also suggest the spleen is a reservoir permissive to high levels of MAYV replication in the absence of innate immunity." Please, show me which of your data demonstrate the spleen as potential viral reservoir in the absence of innate immunity?

4. "The novel AdV-MAYV vaccine approach reported here is the fourth reported vaccine approach against MAYV and is the only MAYV vaccine approach that has shown the ability to cross-protect against both CHIKV and UNAV [8, 10, 31, 60]." Why do you claim to be the fourth reported vaccine candidate?

1. Inactivated MAYV vaccine: 10.1093/milmed/141.3.163

2. DNA MAYV vaccine: 10.1371/journal.pntd.0007042 

3. Attenuated IRES MAYV vaccine: 10.1371/journal.pntd.0002969

4. ??

Reviewer #5: see attached documentation

**Conclusions**

-Are the conclusions supported by the data presented?

-Are the limitations of analysis clearly described?

-Do the authors discuss how these data can be helpful to advance our understanding of the topic under study?

-Is public health relevance addressed?

Reviewer #1: The conclusions are supported by the data. Limitations are described in the discussion, however one point should be discussed in more detail. Why is the cross-protection of UNA less efficient than that of CHIKV. UNA is closer related to MAYV. Please explain or comment.

Reviewer #2: (No Response)

Reviewer #3: -Are the conclusions supported by the data presented? - Yes

-Are the limitations of analysis clearly described? - Partially

-Do the authors discuss how these data can be helpful to advance our understanding of the topic under study? - yes

-Is public health relevance addressed? - yes

Reviewer #4: -Are the conclusions supported by the data presented?

Yes

-Are the limitations of analysis clearly described?

Yes

-Do the authors discuss how these data can be helpful to advance our understanding of the topic under study?

Yes

-Is public health relevance addressed?

Yes

Reviewer #5: see attached documentation

**Editorial and Data Presentation Modifications?**

Reviewer #1: Comments:

Page numbers would have been helpful.

1. Figure 3B. Why are different bands detected with mouse serum compared to CHIKV Abs?

2. Figure 5: The authors should comment on the Eotaxin values. Apparently, AdV-MAYV vaccination inhibits eotaxin compared to the untreated control.

3. Figure 6B: No infection is indicated in the figure legend, but the columns are not visible.

4. Why is the cross-protection of UNA less efficient than that of CHIKV. UNA is closer related to MAYV. Please explain or comment.

Reviewer #2: (No Response)

Reviewer #3: The manuscript by Powers et al. is an outstanding piece of work investigating the efficacy of a non-replicating adenovirus based Mayaro virus vaccine. The authors report on a new

approach using an adenovirus vector that encodes Mayaro virus structural proteins that assemble into non-infectious virus-like particles. They assess vaccine efficacy in immune-competent and -compromised murine models as well as evaluate the cross-protective effects of this vaccine against CHIKV and UNAV. Below are only minor comments meant to improve the manuscript.

1. Use of the phrase urban spread and mentioning urban transmission cycles is a little misleading. Isn’t this incriminated but not proven?

2. Clarify if immunization was intramuscular in the methods here. 

3. Rationale for using the CHIKV Sri Lankan strain and not an American strain.

4. Has this WB procedure been performed previously? I would suspect so since this has been done with AdV-CHIKV. You can cite the original methods and just include changes to the protocol here. Same comment for neutralization assays section.

5. There is no reference to sample size in these mouse experiments

6. Rationale for using the strain of BeArr505411. Is this a genotype L strain?

7. There is no reference for safety studies of this vaccine. Can you discuss the safety of the vaccine in some regard?

8. Figure 6. passive is spelled incorrectly in the legend

9. Authors should describe how the PRNT50s remain comparable between the wt and IFN alpha KO mice

10. The authors don’t discuss the implication of detecting virus in the ipsilateral ankle and its potential impact of arthralgia in the cross-protection studies.

11. Discussion comparing the NT titers with this vaccine and other mayv vaccines would be useful

12. There is a lot of repetition of methods in these results sections which is perfect if the results appear first, but it is not presented this way in the text

13. Address the limitations of your cell-mediated response study

Reviewer #4: Writing suggestions

Introduction

1. "The ability of the virus to infect both Aedes and Culex mosquitos and a wide range of vertebrate hosts

permits both enzootic and urban transmission cycles [2]." 

Actually there is still evolutionary barriers holding MAYV transmission among Vertebrate hosts and Aesde and Culex mosquitos.

I would change the term 'permits'

Please see:

10.4269/ajtmh.2011.11-0359

10.1371/journal.pntd.0006895

2. "However, to date there are no approved vaccines against any alphaviruses. Previous MAYV vaccination attempts have included live-attenuated virus and DNA based vaccines [8-11]." Please include 'inactivated vaccine'

Material and Methods section

Viruses

1. "... The infected cells were rocked continuously..." Suggestion: For viral adsorption, the cells were continuously rocked...;

Quantification of virus tissue burden

1. "...containing confluent monlolayers of Vero cells..." Suggestion: 'Monolayer';

2. "... were fixed with 3.7% formalin diluted in 1x PBS." Suggestion: Remove '1x';

Western Blot Analysis

1. "Membranes were washed and then developed with ThermoFisher Pico". Suggestion: Substitute 'developed' by revealed;

Neutralization Assays

1. "One milliliter of 5% FBS/DMEM/CMC was added to each..." Suggestion: Substitute here and elsewhere '5% FBS/DMEM/CMC' by 'CMC DMEM media containing 5% FBS';

2. "...to each well and the plate was incubated for 48 h." Suggestion: include 'at 37℃ and 5% CO2 atmosphere';

3. "...3.7% formaldehyde to each well for 15 minutes, washed with cold water...". Suggestion: Include 'then the supernatant was removed and the wells were washed with cold water...';

4. "...and stained with 0.2% methyl blue dye for 15 minutes. Plates were washed..." Suggestion: Include: 'The dye was removed and the plates were washed...';

Enzyme-Linked Immunoassay (ELISA)

1. '...30 minutes and 9.4x104 PFU in 100 μl PBS was...' Suggestion: '...MAYV was diluted in PBS to 9.4x10^4 PFU/100uL...';

2. '...blocked for 1 h with 5% milk in 1X TBS with 1%...' Suggestion: Remove '1x';

3. 'Plates were then washed three times with ELISA buffer, blotted dry and developed with OPD substrate.' Suggestion: '...and colorimetric signal was developed...';

Results

Adenovirus Mediated Expression of Mayaro Structural Proteins

1. 'The Mayaro (MAYV)...' Suggestion: The Mayaro virus (MAYV);

2. 'THF-CAR cells were infected with a multiplicity of infection (MOI) equal to 100 PFU/cell...' Suggestion: THF-CAR cells were infected with a MOI equal to 100 PFU/cell;

AdV-MAYV Vaccination Elicits Neutralizing Antibodies and T cell Responses

1. "...that pre-formed CD8+ T cell responses have on alphavirus disease, and therefore, next determined whether AdV-MAYV..." Suggestion: ...we next determined...;

2. "...receptor knockout (IFN⍺R1-/-) mice vaccinated with AdV-MAYV and the cells were plated onto IFN- ELISpot plates..." Suggestion: ...mice vaccinated with AdV-MAYV. The cells were...;

3. "Epitopes in E2 have been identified as prominent targets of early T cell responses in CHIKV infection and support the N-terminus of E2 as an important target of the early adaptive immune response [38, 44]." This phrase should be moved to discussion;

AdV-MAYV Vaccine Elicits Protective Efficacy & Reduces Inflammatory Chemokine Production

1. Title section suggestion: ...Vaccine Elicits Protection and Reduces Inflammatory...;

2. "infectious virus was undetectable in any of the tissues from both groups of AdV-MAYV vaccinated mice but virus was present in all tested tissues from..." Suggestion: ...AdV-MAYV vaccinated mice in contrast all tested...;

3. "In fact, for most chemokines there was no statistical difference in chemokine levels between AdV-MAYV vaccinated..." Suggestion: In fact, for all chemokines...;

Discussion Section

1. "A previous study by Webb et. al., in which mice vaccinated with a CHIKV/IRES vector, as well as CHIKV infected mice, produced MAYV cross-reactive antibodies, implicated shared antigenic epitopes between MAYV and CHIKV [31]." Suggestion: ...produced limitated MAYV cross-reactive antibodies, indicating the existence of common antigenic epitopes...

Figure 4. 

1. "mice were challenged with 104 PFU/ml MAYVBeAr in the right footpad." Suggestion: Change PFU/ml to total PFU;

2. Follow the figure pattern and add the red dots to Fig 4H;

Supplemental Figure 2.

1. Please, double check the objective magnification. They all seems not to be right. Also, please include a properly scale bar information;

Reviewer #5: see attached documentation

**Summary and General Comments**

Reviewer #1: The manuscript describes the very detailed preclinical testing of a MAYV vaccine candidate. This vaccine not only protects mice from MAYV infections but also from CHIKV and UNV, showing that this type of vaccine might have pan-alphavirus protection potential. Since alphaviruses are spreading worldwide this approach should be expanded in the future. Therefore, this manuscript is of high importance for the field.

Reviewer #2: Major issues

1) A comprehensive paper on a disease that is very minor with only 30-100 cases per annum (Nat Rev Rheumatol. 2012 May 8;8(7):420-9). This is not enough to justify a vaccine, either commercially or on public health grounds as most of these infections are self resolving. That the virus might remerge to be more of a problem is possible (e.g. Pathogens. 2020 Sep 8;9(9):738; Int J Infect Dis. 2020 Mar;92:253-258), but not reality at this stage. The paper should make these issues clear e.g. case numbers, market size etc and explain that MAYV is currently not a viable target for a vaccine. 

2) The notion that anyone would make a MAYV vaccine in order to target CHIKV is plainly extremely unlikely given that several CHIKV vaccines are already at advanced stages of development. As outlined in ref 33, the magnitude of cross protection needs to be addressed or at least alluded to. If you need 10x more vaccine to achieve cross protection against CHIKV than you do to get protection against MAYV, regulators are unlikely to permit use of 10x the effective dose for the primary target, with cost of goods also 10 times higher. This cross-protection study is thus an academic issue and should not be presented as some kind of conceivable reality with any actual real world utility. Arguing this is the only approach to show cross protection is somewhat disingenuous as few would warrant pursuing this as a valuable target of study. By simply upping the dose to beyond what is needed for protection against the primary target, cross-protection is achieved. Most would choose the lowest dose of vaccine to afford protection against the primary target (thereby not achieving cross-protection). Thus protection has less to do with the vaccine modality and is more depended on vaccine dose. Protection against alphaviruses requires neutralisation titers as low as 1 in 5, in this study titres are >1 in 100. The paper needs to be rewritten to remove this erroneous logic/rationale. 

3) Statistics has some problems as parametric tests are used throughout when clearly some of the data is non-parametric. Fig. 5 RANTES – no way do these data have equal variance. Fig 5, Eotaxin – clearly highly skewed data i.e. not normal distribution. Non parametric tests should be used.

Minor issues

1) The discussion on IgG3 and E2 is not so compelling. Interplay between IgM and IgG2b needs a reference and how this is supposed to relate to IgG3 is unclear. Suggest consider deletion.

2) The First part of the Discussion, up to “In this report” is introduction and needs to be moved. 

3) Should make clear 6J vs 6N mice (Lab Anim Res. 2017 Jun;33(2):119-123).

4) Ref 38 is in RAG mice which is somewhat out of context here given the very clear data that CD8 T cells have no protective role under normal circumstances (Front Immunol. 2020 Feb 21;11:287).

5) Joint and muscle reservoirs needs a reference and is misleading – these may be sites of replication (PLoS Pathog. 2017 Dec 27;13(12):e1006788) but reservoir implies something quite different. 

6) “levels than infected mice.” Seems to have some words missing higher than what?

7) Fig.2B PRNTs might have some SDs added.

Several statements in the paper are incorrect

8) “No approved vaccines against any alphavirus”. This is incorrect a Getah vaccine is commercial available in Japan from Nisseiken (https://www.jp-nisseiken.co.jp/en/products/pdf/horse/JE_GETA_en.pdf).

9) “effective treatments” - NSAIDs are unquestionably an effective treatment, and are widely used. Like all treatments for anything, they are not 100% effective, but they are nevertheless effective at ameliorating arthropathy. 

10) The claims regarding myalgia and arthralgia cannot be made as these terms are defined as pain in joints and pain in muscle, and there is no way pain can be measured in these mouse models.

Reviewer #3: All comments are include above

Reviewer #4: In this Manuscript Power JM et al., describes a potential adenovirus-based vaccine candidate for MAYV with potential cross-protective effectivenes to other alphaviruses, including Chikungunya and Una virus. They utilized immunocompetent and IFNaR1-/- mice to evaluate the protective efficacy of immune response developement regarding viral control, neutralizing and protective antibodies development, cytokines production control, disease score and lethality upon challenge. 

This work represents an innovative vaccine development approach for emergent viral diseases caused by MAYV, CHIKV and UNAV, which are a world public health concern specially in undeveloped countries. However, at the time of manuscript revision, it has been published elsewhere a manuscript that utilized the same vaccine strategy, describing an Adenoviral-Vectored MAYV and CHIKV vaccine candidate, which are able to confer partial cross-protection against viral infection in mice (10.3389/fimmu.2020.591885). Very similarly, their findings demonstrate that AdV-MAYV vaccine provides full protection (reduction of viremia, weight loss and foot swelling) against MAYV challenge with increased neutralizing antibodies production, which partially cross-protect against CHIKV. Please, I would suggest that these findinds should be commented and discussed in the manuscript.

Reviewer #5: see attached documentation

PLOS authors have the option to publish the peer review history of their article (what does this mean?). If published, this will include your full peer review and any attached files.

Reviewer #1: No

Reviewer #2: No

Reviewer #3: No

Reviewer #4: Yes: Marcilio Jorge Fumagalli

Reviewer #5: No
---

## [Decision Letter · Decision Letter 1]

15 Mar 2021

Dear Dr. Streblow,

We are pleased to inform you that your manuscript 'Non-Replicating Adenovirus Based Mayaro Virus Vaccine Elicits Protective Immune Responses and Cross Protects Against Other Alphaviruses' has been provisionally accepted for publication in PLOS Neglected Tropical Diseases.

Best regards,

Ran Wang, Ph.D., M.D.

Associate Editor

Francis Jiggins

Deputy Editor

Reviewer's Responses to Questions

**Key Review Criteria Required for Acceptance?**

**Methods**

-Are the objectives of the study clearly articulated with a clear testable hypothesis stated?

-Is the study design appropriate to address the stated objectives?

-Is the population clearly described and appropriate for the hypothesis being tested?

-Is the sample size sufficient to ensure adequate power to address the hypothesis being tested?

-Were correct statistical analysis used to support conclusions?

-Are there concerns about ethical or regulatory requirements being met?

Reviewer #1: The methods are adequately chosen and described in detail. The conclusions drawn by the authors are well supported be the data.

The animal experiments met regulators requirements.

Reviewer #3: -Are the objectives of the study clearly articulated with a clear testable hypothesis stated? - yes

-Is the study design appropriate to address the stated objectives? - yes

-Is the population clearly described and appropriate for the hypothesis being tested? - yes

-Is the sample size sufficient to ensure adequate power to address the hypothesis being tested? - yes

-Were correct statistical analysis used to support conclusions? - yes

-Are there concerns about ethical or regulatory requirements being met? - no

Reviewer #4: 1. Adenovirus vaccine Vector

For amplification of MAYV gene, before the PCR product amplification, you need the cDNA production, since you are working with an RNA virus. Please include this comment.

"We apologize to the reviewer that it was confusing about whether we produced cDNA prior to detection. This has been modified for clarity in the materials and methods section."

Please, include how was the cDNA produced before MAYV structural genes amplification and cloning.

2. Mouse experiments

For viral feet challenge, why did you use 10-log reduced viral load for CHIKV (10^3 PFU) compared to MAYV and UNAV?

"While we’ve been using CHIKV at 1e3 PFU for a while, we used data mined from the literature to establish the dosage for MAYV and UNAV."

Please include the references.

3. Quantification of virus tissue burden

It is mentioned that you used 500uL of PBS for tissue harvest. If you used the same PBS volume for all collected tissue samples, I assume they would present different tissue concentrations, depending on sample size/mass. How did you normalized your PFU/qRT-PCR and cytokines results? I see that your tissue viral loads (PFU/mL and copies/mL) and cytokines quantification (pg/mL) are all in mL of tissue homogenate.

"Yes, all tissues were collected into the same volume of PBS and then homogenized by bead beating. These samples were used for cytokine, PFU, and vRNA assessments. We typically do not compare between tissue types only within a specific type. Thus, the samples are controlled within specific tissue type."

Unfortunately, you are comparing Cytokines concentration and viral titers among the same sample specimen groups and between different groups (Please see Figure 5; Figure 8E, Figure 10 E-H). Since you are setting you data output as a concentration (pg/mL and PFU/mL), it is extremely necessary to indicate how did you normalize your sample concentration. If they were all processed in different tissue concentrations, this would directly affect your data readout if not normalized.

4. qRT-PCR

The MAYV reverse primer does not anneal in a reverse way to the target region. The complementary reverse sequence would be '5- CTTCCAGGCTGCCCGGCACCAT -3'. Please, double-check the sequence;

"We apologize the sequence was reverse/complement and has been fixed."

Again, the reverse sequence still not annealing to any target region on MAYV TRVL 4675. Please, provide the correct sequence.

Reviewer #5: Methods and resources clearly presented this study.

**Results**

-Does the analysis presented match the analysis plan?

-Are the results clearly and completely presented?

-Are the figures (Tables, Images) of sufficient quality for clarity?

Reviewer #1: The experiment are presented in a logic order and very well supported the conclusions the author stated. Minor modifications of the Figures were adjusted and fullfil the requirements now.

Reviewer #3: -Does the analysis presented match the analysis plan? - NA

-Are the results clearly and completely presented? - yes

-Are the figures (Tables, Images) of sufficient quality for clarity? - yes

Reviewer #4: (No Response)

Reviewer #5: Yes

**Conclusions**

-Are the conclusions supported by the data presented?

-Are the limitations of analysis clearly described?

-Do the authors discuss how these data can be helpful to advance our understanding of the topic under study?

-Is public health relevance addressed?

Reviewer #1: The conclusions are supported by the data. Limitations are described in the discussion, and the cross-protection of UNA was sufficiently explained.

Reviewer #3: -Are the conclusions supported by the data presented? - yes

-Are the limitations of analysis clearly described? - yes

-Do the authors discuss how these data can be helpful to advance our understanding of the topic under study? - yes

-Is public health relevance addressed? - yes

Reviewer #4: (No Response)

Reviewer #5: The scientific premise of this study is supported by the referenced literature.

**Editorial and Data Presentation Modifications?**

Reviewer #1: (No Response)

Reviewer #3: The authors have addressed my concerns.

Reviewer #4: (No Response)

Reviewer #5: None

**Summary and General Comments**

Reviewer #1: The revised manuscript is now acceptable for publication.

Reviewer #3: (No Response)

Reviewer #4: The manuscript by Powert JM et al., describes a very interesting adenovirus-based vaccine candidate for MAYV. The manuscript is well described and the immunological outcomes have been well evaluated, including cross-protective propertites against other closely related alphaviurses. As I wrote before, this work represents an innovative vaccine development approach for emergent viral diseases caused by MAYV, CHIKV and UNAV.

Almost all my inquiries have been addressed, but at this point I would recommend acceptance of this manuscript.

Reviewer #5: (No Response)

PLOS authors have the option to publish the peer review history of their article (what does this mean?). If published, this will include your full peer review and any attached files.

Reviewer #1: No

Reviewer #3: No

Reviewer #4: **Yes: **Marcilio Jorge Fumagalli

Reviewer #5: No

---

## [Editor Report · Acceptance letter]

26 Mar 2021

Dear Dr. Streblow,

We are delighted to inform you that your manuscript, "Non-Replicating Adenovirus Based Mayaro Virus Vaccine Elicits Protective Immune Responses and Cross Protects Against Other Alphaviruses," has been formally accepted for publication in PLOS Neglected Tropical Diseases.

Best regards,

Shaden Kamhawi

co-Editor-in-Chief

Paul Brindley

co-Editor-in-Chief
